# Position: Reasoning After Perception Means Reasoning Without Vision

**Hongcheng Gao** [* 1]  **Zihao Huang** [* 2]  **Jingyi Tang** [* 3]  **Lin Xu** [1]  **Xinhao Li** [4]  **Xinhao Li** [5]  **Yue Liu** [6]  **Minhua Lin**
**Xinlong Yang** [3]  **Taihang Hu** [7]  **Ge Wu** [7]  **Baolong Bi** [8]  **Hongyu Chen** [8]  **Zhiqi Huang** [3]  **Wentao Zhang** [3 9]

## Abstract

A common belief in multimodal research is that the perceptual weaknesses of vision–language models can be compensated by stronger language reasoning (e.g., chain-of-thought, in-context learning, or external tools). We challenge this assumption. We argue that for a broad class of visual tasks hard to specify in language, failures stem from a structural fatality where the temporal decision of *when* to reason strictly dictates the spatial constraint of *where* reasoning takes place. When visual reasoning is deferred to language generation, current architectures do not merely delay computation; they displace it from the continuous visual representation to a discrete textual space. Consequently, the sequential "Perception-then-Reasoning" paradigm degenerates perception into a passive, one-off feature encoding process, rendering it functionally equivalent to "Reasoning-in-Text-Space", where task-critical spatial signals are collapsed before reasoning begins. We substantiate this claim with the Turing Eye Test (TET): tasks that must be resolved in *visual space* and are hard to verbalize; results show text-only reasoning cannot remedy these perceptual failures. Our findings suggest rethinking the architectural divide: shifting from reasoning *about* perception to reasoning *within* perception. This facilitates actively reasoning-driven perception that operates directly on pixel-level visual representations, rather than within a collapsed textual space.

## 1. Introduction

Multimodal Large Language Models (MLLMs) exhibit striking linguistic fluency and often give the impression of com-

*Equal contribution [1]Tsinghua University [2]BIT [3]Peking University [4]Nanjing University [5]BUPT [6]National University of Singapore [7]Nankai University [8]UCAS [9]Zhongguancun Academy. Correspondence to: Hongcheng Gao <gaohongcheng2000@gmail.com>, Wentao Zhang <wentao.zhang@pku.edu.cn>.

*Proceedings of the 43rd International Conference on Machine Learning*, Seoul, South Korea. PMLR 306, 2026. Copyright 2026 by the author(s).

prehensive visual understanding (Chen et al., 2024b; Bai et al., 2025a; Team, 2025e; Guo et al., 2025; Team, 2025c; An et al., 2025b). Yet they still fail on tasks that are *conceptually simple* but *hard to specify precisely in text*: deciding whether two segments intersect, whether two shapes are congruent, identifying which regions are connected, as well as spotting differences between similar images (Tong et al., 2024; Rudman et al., 2025; Chen et al., 2024a; Bai et al., 2025c). These questions are visually straightforward given complete images, but brittle under language-only specifications. This gap suggests a structural mismatch: current systems are excellent at *talking about* images, but unreliable at *preserving and querying* the fine-grained spatial evidence such decisions require (Wang et al., 2025e; Ma et al., 2026; Wu et al., 2024; Cao et al., 2025).

A common response attributes these failures to insufficient downstream reasoning, assuming that scaling language-side deliberation (e.g., CoT (Wei et al., 2022; Zhang et al., 2024b)) can resolve ambiguities. **We challenge this assumption.** For a broad class of geometry-, topology-, and counting-driven tasks, we argue that the limiting factor is not the reasoning depth, but the accessibility of the *right visual evidence* (Tong et al., 2024). This stems from a structural fatality where the **temporal decision of *when* to reason strictly dictates the spatial constraint of *where* reasoning takes place.** By deferring visual reasoning to language generation, current architectures **displace computation** from the continuous visual manifold to a discrete textual space, degenerating perception into a **passive, one-off feature encoding process** that collapses task-critical signals before reasoning begins.

> **Position**
>
> The prevailing "Perception-then-Reasoning" paradigm creates a structural bottleneck: deferring reasoning to the decoder constrains it to a discrete, text-aligned space where task-critical signals have already collapsed. We advocate for **reasoning *within* perception**—operating on pixel-level representations rather than collapsed textual proxies.

Our central claim is that **the temporal decision of *when* to reason dictates *where* reasoning can occur.** In most MLLMs, decisive computation is deferred until language de-

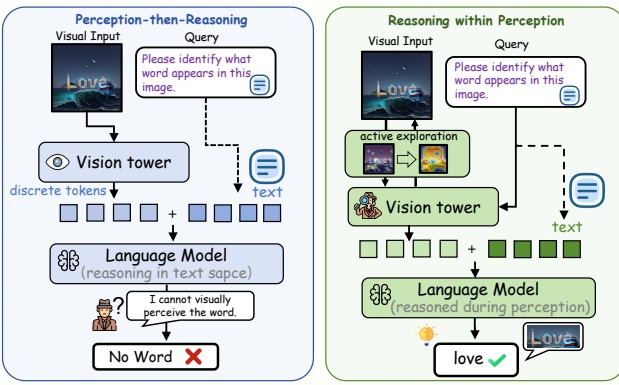

*Figure 1.* **(Left)** The mainstream *Perception-then-Reasoning* paradigm: visual evidence is passively encoded into semantic signals, leading to *Information Collapse.* **(Right)** The proposed *Reasoning within Perception* paradigm: reasoning is executed actively on the original image before semantic compression.

coding, after the visual stream has been compressed into a text-facing interface (e.g., a limited set of visual tokens, pooled embeddings, or cross-attended summaries) (Bai et al., 2025a; An et al., 2025b). Consequently, the sequential "Perception-then-Reasoning" paradigm **degenerates perception into a passive, one-off feature encoding process**, rendering it functionally equivalent to "Reasoning-in-Text-Space." This design is particularly detrimental for tasks that are hard to specify in text yet easy to decide from pixels: the distinctions required for correctness are not reliably available in the text-aligned representation at reasoning time. Thus, increasing textual deliberation can improve coherence and self-consistency while still failing to recover signals that were not preserved through the bottleneck (Tang et al., 2025; Qu et al., 2025).

We provide diagnostic evidence using the **Turing Eye Test (TET)**, a probe designed to reduce the usefulness of linguistic shortcuts and isolate perceptual failures from high-level semantics. Across contemporary MLLMs, we observe that common inference-time interventions—including Chain-of-Thought and in-context learning—often yield limited improvements on hard-to-verbalize visual cases, aligning with the hypothesis that scaling textual deliberation cannot recover information lost during passive encoding.

Consequently, we argue for a shift from reasoning *about* perception to **reasoning *within* perception**. This position implies two architectural realignments:

**1. When and Where to Think: Visual Reasoning During Perception.** Reasoning that depends on geometry and spatial relations should be executed *before* visual evidence is funneled into a text-aligned bottleneck, so that deliberation occurs in a visual space that supports perceptual operations.

**2. What to Preserve: Pixel–Semantic Dual Requirements.** Visual representations should facilitate *active perception*. This requires not only semantic alignment for com-

munication but also the preservation of pixel-level fidelity, allowing high-level semantics to query and steer the visual encoding process to resolve ambiguities and lost information that are invisible in the *discrete text space*.

In summary, we argue that persistent visual failures stem from *representational access*—task-critical spatial information is discarded before reasoning begins—not insufficient reasoning capacity. We develop this claim through a **structural critique** of the mismatch between visual tasks and text-space reasoning (Sec. 3), a **theoretical analysis** formalizing information collapse in vision-to-language projection (Sec. 4), **empirical validation** via TET (Sec. 5), and a **proposed framework** for reasoning within perception (Sec. 6).

## 2. Background: Architectural Paradigms

Before presenting our critique, we establish a taxonomy of existing approaches based on *when* and *where* visual reasoning occurs. Table 1 summarizes key architectural paradigms and their implicit assumptions.

**Perception-then-Reasoning.** The dominant paradigm treats vision as preprocessing, compressing images into discrete tokens consumed by LLMs (Liu et al., 2023; Dai et al., 2023; Alayrac et al., 2022). Reasoning occurs entirely post-compression in text space. Variations include static projection (direct mapping from visual features to text embeddings) (Liu et al., 2024), enhanced inference (CoT, ICL, or sampling strategies applied during text generation) (Wang et al., 2025c; Chen et al., 2025c; Aissi et al., 2025; Huang et al., 2025), and tool augmentation (external modules invoked during generation) (An et al., 2025a; Surís et al., 2023; Wang et al., 2025a).

**Reasoning Within Perception.** Recent works explore performing computation before or during visual encoding, including visual sketching approaches that enable iterative refinement in latent visual space (Zhang et al., 2025a; Chung et al., 2025), and unified tokenization that treats vision and language symmetrically (Xie et al., 2025; Chen et al., 2025b). Our position advocates for this direction but identifies persistent limitations even in these approaches, particularly semantic bias inherited from text-aligned pretraining.

Table 2 situates TET relative to existing multimodal benchmarks. Many general MLLM benchmarks emphasize semantic correctness or coarse QA accuracy, so strong performance can still be achieved through language priors or distributional shortcuts. TET instead suppresses linguistic shortcuts and probes whether fine-grained visual evidence remains accessible after vision-language projection.

## 3. Structural Critique: Description ≠ Seeing

Current MLLM paradigms have achieved remarkable success in high-level semantic understanding tasks (Yue et al.,

*Table 1.* **Taxonomy of Visual Reasoning Paradigms.** We categorize approaches by when reasoning occurs, where computation takes place, and the underlying perception mode. Our proposal advocates for iterative refinement within the naive visual space.

| Paradigm | When | Where | Perception Mode | Representative Work |
|---|---|---|---|---|
| *Current Mainstream: Perception-then-Reasoning* | | | | |
| CoT Enhancement | Post-LLM Decode | Text Space | Static Semantic Perception | Gemini (Team, 2025b) QVQ (Team, 2024) Kimi k1.5 (Team, 2025d) Qwen3-VL (Bai et al., 2025a) |
| Tool Augmentation | Post-LLM Decode | Text Space | Multi-turn Static Semantic Perception | ViperGPT (Surís et al., 2023) Mini-o3 (Lai et al., 2025) DeepEyes (Zheng et al., 2025) VTool-R1 (Wu et al., 2025c) OpenThinkIMG (Su et al., 2025) |
| *Emerging: Toward Reasoning Within Perception* | | | | |
| Visual Sketching | Post-VT | Latent Space | Static Semantic Perception | LatentSketchpad (Zhang et al., 2025a) MVoT (Li et al., 2025a) |
| Unified Model | Post-LLM Decode | Visual Space | Hybrid Semantic-Texture Perception | Show-o2 (Xie et al., 2025) Janus-Pro (Chen et al., 2025b) Bagel (Deng et al., 2025) |
| **Our Proposal** | **Pre-VT** | **Naive Visual Space** | **Active Reasoning within Perception** | **Active Visual Querying** |

*Table 2.* **Landscape of Visual Reasoning Benchmarks. Type I & II** benchmarks possess high verbalizability, allowing models to reduce visual tasks to text-based deduction. **Type III** benchmarks expose perceptual bottlenecks where linguistic shortcuts fail. The last column highlights relevance to our central thesis.

| Benchmark | Core Task Focus | Verbalizability | Reasoning Substrate | Primary Bottleneck | Relevance to Thesis |
|---|---|---|---|---|---|
| *Type I: Semantic & Knowledge-Intensive* | | | | | |
| MMMU (Yue et al., 2024) | College-level exams | High | World Knowledge | Knowledge retrieval | Success stems from text reasoning, not visual understanding. |
| MathVista (Lu et al., 2024) | Math problem solving | Medium | Text/Formulas | Symbolic manipulation | |
| ScienceQA (Lu et al., 2022) | Multimodal science Q&A | High | World Knowledge | Domain knowledge | |
| OmniDocBench (Ouyang et al., 2025) | Document parsing | High | Text & Layout | Structure recognition | |
| *Type II: Abstract & Symbolic Logic* | | | | | |
| LogicVista (Xiao et al., 2024) | Logical reasoning | Medium | Discrete Symbols | Rule induction | Tests discrete rule-following, bypassing visual bottleneck. |
| ARC-AGI-2 (Chollet et al., 2026) | Grid pattern abstraction | High | Discrete Grid State | Program synthesis | |
| *Type III: Low-Level Perception* | | | | | |
| MMVP (Tong et al., 2024) | Visual fundamentals | Low | Visual Features | Vision encoder | Exposes perceptual bottleneck, but reasoning ability remains untested. |
| Blink (Fu et al., 2024) | Core perception | Low | Visual Features | Rapid perception | |
| BabyVision (Chen et al., 2026) | Pre-linguistic vision | Low | Visual Primitives | Visual primitives | |
| HiddenInPlainSight (Fu et al., 2025) | Visual grounding | Low | Visual Features | LLM integration | |
| **TET (Ours)** | **Fine-grained reasoning** | **Low** | **Continuous Manifold** | **Visual reasoning** | **Proves text reasoning failed.** |

2024; Lu et al., 2024), yet they suffer from a fundamental architectural misalignment. We argue that the community has been optimizing for the wrong objective: **we are teaching models to "describe" the world, not to "see" it.**

### 3.1. The "Where to Think" Fallacy

The dominant architecture treats vision as a preprocessing step, compressing images into discrete tokens to be processed by an LLM. This forces all reasoning to occur in the *textual latent space*. However, visual reasoning is inherently different from linguistic reasoning. Tasks such as mental rotation, trajectory tracking, and spatial geometry rely on continuous transformations, not discrete symbol manipulation. Forcing an LLM to simulate continuous geometric changes using discrete text tokens is computationally inefficient and fundamentally flawed. Table 3 presents representative examples illustrating this fundamental mismatch between task requirements and architectural support.

The bottleneck is not the LLM's reasoning capability, but the representation itself. We argue that visual reasoning must occur in the *native image space* (or a continuous visual manifold), not in the text space. The question of "where to think" must be answered by shifting computation back to the visual domain.

### 3.2. The "What to See" Deficit

Current visual encoders (e.g., CLIP) are trained to align with text (Radford et al., 2021), which biases them towards *semantic* features while discarding task-critical *spatial signals* (Zhang et al., 2025b). They can identify "a cat on a table" but struggle to determine whether two line segments intersect or count disconnected regions—tasks that are visually straightforward yet hard to specify in text.

This creates a representational disconnect: the model knows *what* objects are present but cannot resolve *how* they relate spatially. We hypothesize that the projection from visual to textual space acts as a filter governed by verbalizability: information that can be easily described in language is preserved, while information that resists linguistic specification is systematically collapsed. Table 4 categorizes information

*Table 3.* **Task Requirements vs. Architectural Support.** Examples illustrating disparity between the continuous nature of visual tasks and the discrete representations in current architectures.

| Task Category | Required Operation | Current Support |
|---|---|---|
| Mental Rotation (Tong et al., 2024; Huang et al., 2026) | Continuous transform | Discrete tokens |
| Occlusion Reasoning (Wang et al., 2025b; Liu et al., 2025b) | Depth ordering | Flattened features |
| Spatial Counting (Daxberger et al., 2025; Wu et al., 2025a) | Pixel-level grouping | Semantic categories |
| Trajectory Prediction (Liu et al., 2025a; Kim et al., 2025) | Temporal dynamics | Static snapshots |
| Geometric Measurement (Li et al., 2024; Xu et al., 2025b) | Metric distances | Categorical labels |
| Fine-grainedw Recognition (He et al., 2025; Kuchibhotla et al., 2025) | Stroke-level detail | Patch-level semantics |

types along this axis; our experimental results in Sec. 5 provide empirical validation.

### 3.3. The Sequential Trap vs. Perception as Reasoning

The prevailing pipeline of MLLMs follows a rigid sequence: *Perception → Alignment → Reasoning*. This assumes that perception is a one-off extraction process that finishes before reasoning begins. We contend that this separation is artificial. In biological systems, perception and reasoning are intertwined; we reason *while* we perceive (Marr, 1982). Crucially, the standard practice of "freezing" the perception

*Table 4.* **Hypothesized Semantic Bias of Vision-Language Projection.** Verbalizability determines information survival.

| Information Type | Easy to Verbalize? | Prediction |
|---|---|---|
| Semantic categories | Yes | Preserved |
| Coarse spatial layout | Yes | Preserved |
| Pixel-level fidelity | No | Collapsed |
| Geometric relations | No | Collapsed |
| Topological structure | No | Collapsed |
| Stroke-level details | No | Collapsed |

module reduces vision to a passive, one-off encoding step. This creates a structural barrier: the reasoning engine is unable to leverage linguistic inputs to actively re-examine the visual data for task-relevant details. Since this static representation inevitably captures less information than the raw visual input, critical fine-grained signals are often discarded before reasoning even begins. To move forward, we must abandon the "Perception *then* Reasoning" paradigm in favor of **"Perception *as* Reasoning"** where the model actively reasons within the visual signal, iteratively refining its understanding at both the semantic and pixel levels.

### 3.4. External Tool Invocation Is Not Native Seeing

To bridge the gap between human thought processes and pure text-based reasoning, the community has attempted to integrate visual tools into the reasoning pipeline. Exemplified by models like o3 (OpenAI, 2025), tool-augmented methods (Lai et al., 2025; Zheng et al., 2025) enable models to engage in *multi-round visual evidence* by iteratively invoking external utilities—ranging from code interpreters to plotting libraries—to process or reconstruct visual data.

However, these methods fundamentally remain limited by a loose coupling between cognition and perception. While the

model can direct its "hands" (external tools) to execute complex visual operations, the *critical reasoning steps* are still strictly performed within the "brain" (the LLM) in a purely textual manner. The reasoning engine merely consumes the discrete outputs of these tools without direct access to the continuous visual process. This is akin to handing a blind poet a more capable pen; he remains without eyes.

## 4. Theoretical Analysis

We now formalize the limitations of current MLLM paradigms and derive a theoretical characterization of the error introduced by text-space reasoning.

### 4.1. Problem Formalization and Notation

Let $\mathcal{X} \subseteq \mathbb{R}^{H \times W \times 3}$ denote the continuous image manifold, and let $y \in \mathcal{Y}$ represent the task label. The architecture comprises three primary functional components:

**1. Vision Encoder:** $f_\theta : \mathcal{X} \to \mathcal{Z} \subseteq \mathbb{R}^d$, which maps the image manifold to a continuous visual feature space $\mathcal{Z}$.

**2. Modality Projection:** $g_\phi : \mathcal{Z} \to \mathcal{E} \subseteq \mathbb{R}^D$, a mapping from visual features to the text-aligned latent space $\mathcal{E}$ of the Large Language Model. Note that $d$ and $D$ may differ; this projection is optimized for semantic alignment rather than geometric fidelity.

**3. Task Head:** $h_\psi : \mathcal{E} \to \hat{\mathcal{Y}}$, which generates the final prediction. The training objective is to minimize the empirical risk:

$$\hat{\mathcal{R}}(\theta, \phi, \psi) = \frac{1}{N} \sum_{i=1}^{N} \ell(h_\psi \circ g_\phi \circ f_\theta(x_i), y_i). \quad (1)$$

In the following analysis, we focus on per-sample error to isolate the structural bottleneck.

### 4.2. The Information Collapse Bound

To characterize the representational bottleneck, we introduce two key concepts:

**Task-Relevant Representation.** For a task with label $y$, let $z^* \in \mathcal{Z}$ denote the ideal visual representation—one that preserves exactly the geometric and spatial information required to predict $y$.

**Reconstruction Mapping.** To rigorously measure information loss across heterogeneous spaces ($\mathcal{Z}$ and $\mathcal{E}$ with potentially different dimensions), let $g^\dagger : \mathcal{E} \to \mathcal{Z}$ denote the optimal reconstruction mapping that attempts to recover visual features from text-aligned embeddings. The *effective* visual information available to the reasoning module is then $\tilde{z} = g^\dagger(g_\phi(f_\theta(x)))$. Applying the triangle inequality in the visual feature space $\mathcal{Z}$:

$$\|z^* - \tilde{z}\|_\mathcal{Z} \leq \underbrace{\|z^* - f_\theta(x)\|_\mathcal{Z}}_{\varepsilon_{\mathrm{enc}}} + \underbrace{\|f_\theta(x) - g^\dagger(g_\phi(f_\theta(x)))\|_\mathcal{Z}}_{\varepsilon_{\mathrm{proj}}}$$

(2)

where $\| \cdot \|_\mathcal{Z}$ denotes a task-appropriate norm on the visual feature space. Here: (1) $\varepsilon_{\mathrm{enc}}$ quantifies the **encoding gap**: the discrepancy between the vision encoder's output and the task-relevant representation. (2) $\varepsilon_{\mathrm{proj}}$ quantifies the **projection gap** (reconstruction error): the information irreversibly lost when visual features are projected into the text-aligned space. This measures the non-invertibility of $g_\phi$.

Under the assumption that the task function is Lipschitz continuous with respect to the visual representation (i.e., small changes in visual state should yield proportionally small changes in the output), the prediction error is bounded by the representation error:

$$\|\hat{y} - y\| \leq L \cdot \|z^* - \tilde{z}\|_\mathcal{Z} \leq L \cdot (\varepsilon_{\mathrm{enc}} + \varepsilon_{\mathrm{proj}}) \quad (3)$$

for some task-dependent constant $L > 0$. While Eq. 3 establishes an upper bound, the existence of a non-zero *lower bound* is guaranteed by the failure of injectivity described below: if distinct visual states collapse to the same $\tilde{z}$, the model cannot simultaneously predict distinct labels correctly.

**The Projection Gap is Non-Vanishing.** A critical observation is that $\varepsilon_{\mathrm{proj}} > 0$ for tasks requiring fine-grained spatial reasoning. The projection $g_\phi$ is *semantically contractive*: it maps spatially distinct but semantically similar visual states to nearby embeddings. Formally, for two images $x_1, x_2$ with identical semantic content but different geometric configurations, the distance in $\mathcal{E}$ is insensitive to geometric variations compared to distance in $\mathcal{Z}$:

$$\|g_\phi(f_\theta(x_1)) - g_\phi(f_\theta(x_2))\|_\mathcal{E} \ll \|f_\theta(x_1) - f_\theta(x_2)\|_\mathcal{Z} \quad (4)$$

This contraction implies that perfect reconstruction is impossible: the mapping $g_\phi \circ g^\dagger$ cannot be the identity on $\mathcal{Z}$ for the subspace encoding geometric details.

**Proposition 4.1** (Information Collapse). *Let $z^*$ encode geometric or topological properties, and let $g_\phi$ be semantically contractive (collapsing geometrically distinct but semantically equivalent inputs). Then: (1) the projection gap $\varepsilon_{\mathrm{proj}} = \|f_\theta(x) - g^\dagger(g_\phi(f_\theta(x)))\|_\mathcal{Z} > 0$ is strictly positive, and (2) the resulting prediction error admits a non-zero lower bound irreducible by text-space reasoning $h_\psi$ alone.*

**Interpretation.** Proposition 4.1 formalizes the "Blind Poet" paradox: when reasoning is confined to the text-aligned space $\mathcal{E}$, the model can only access information that survives the lossy projection $g_\phi$. Scaling text-side computation cannot recover geometric details that were collapsed during projection. **No amount of eloquence can compensate for blindness.**

# 5. Diagnostic Evidence: The Turing Eye Test

To empirically validate our theoretical claims, we introduce the **Turing Eye Test (TET)**—a diagnostic benchmark designed to isolate perceptual failures from high-level semantics. Unlike benchmarks that conflate visual perception with world knowledge or linguistic reasoning, TET specifically targets tasks that are *easy to verify from pixels but hard to specify in text*.

## 5.1. Task Design Principles

TET comprises four task families targeting distinct aspects of fine-grained visual perception: **HiddenText** (150 images)—scale-variant images where text resolves into detailed scenes when magnified; **3DCaptcha** (150 images)—characters with 3D perspective distortion and curved surfaces; **ColorBlind** (150 images)—Ishihara-style tests (Ishihara, 1951) with chromatically confounding elements; and **ChineseLigatures** (40 images)—fused glyphs synthesized from multiple Chinese characters. Representative examples are shown in Fig. 2 and generation details are provided in Appendix A.

## 5.2. Main Results: Collective Blindness

We provide the specific details of our evaluation in Appendix.A.2 and report our evaluation results in Table 5. Despite impressive capabilities on standard multimodal benchmarks (Yue et al., 2025; Lu et al., 2024), **all models exhibit near-zero performance on TET tasks**. This may be because the information passively received by the model is irrelevant to the query. We present attention-map analyses of the Qwen2.5-VL series on HiddenText in Fig. 4, with additional analyses in Appendix C and post-fine-tuning analyses in Appendix D. This failure pattern is consistent with our theoretical analysis: continuous spatial-visual information is forcibly collapsed into discrete textual space, and this dimensional collapse results in catastrophic failure on tasks requiring fine-grained spatial reasoning. Representative response examples are provided in Fig. 15.

## 5.3. Can Enhanced Inference Overcome the Bottleneck?

We investigate whether providing domain-specific exemplars enables models to learn relevant perceptual patterns. Table 6 shows results with 3-shot **in-context learning**. The

*Table 5.* **TET Main Results.** Pass@1 and pass@32 accuracy (%) across 15 state-of-the-art MLLMs. Near-zero performance across all models validates **our claim that current architectures fundamentally lack pixel-level reasoning capability**.

| Category | Model | HiddenText | | 3DCaptcha | | ColorBlind | | ChineseLigatures | | Average | |
|---|---|---|---|---|---|---|---|---|---|---|---|
| | | Pass@1 | Pass@32 | Pass@1 | Pass@32 | Pass@1 | Pass@32 | Pass@1 | Pass@32 | Pass@1 | Pass@32 |
| Unified Multimodal | Show-o2 (Xie et al., 2025) | 0 | 0 | 0 | 0 | 0 | 0 | 0 | 0 | 0 | 0 |
| | Bagel (Deng et al., 2025) | 0 | 0 | 0 | 0 | 0 | 0 | 0 | 0 | 0 | 0 |
| | Janus-Pro (Chen et al., 2025b) | 0 | 0 | 0 | 0 | 0 | 0 | 0 | 0 | 0 | 0 |
| Closed-Source APIs | Claude 4-Sonnet (Anthropic, 2025) | 0 | 0 | 0 | 0 | 0 | 0 | 0 | 0 | 0 | 0 |
| | Gemini 2.5 Pro (Team, 2025b) | 0 | 0 | 0 | 0 | 0 | 0 | 2.5 | 5 | 0.2 | 0.4 |
| | OpenAI o1 (Jaech et al., 2024) | 0 | 0 | 0 | 0 | 0 | 1.33 | 0 | 0 | 0 | 0.4 |
| | Seed-1.6 (Team, 2025a) | 0 | 0 | 0 | 0 | 0 | 0 | 2.5 | 2.5 | 0.2 | 0.2 |
| Open-Source Models | Qwen2.5VL-72B (Bai et al., 2025b) | 0 | 0 | 0 | 0 | 0 | 0 | 0 | 0 | 0 | 0 |
| | Qwen2.5VL-7B (Bai et al., 2025b) | 0 | 0.67 | 0 | 0 | 0 | 0 | 0 | 2.5 | 0 | 0.4 |
| | Qwen2.5-Omni-7B (Xu et al., 2025a) | 0 | 0 | 0 | 0 | 0 | 0 | 0 | 2.5 | 0 | 0.2 |
| | QVQ-72B (Team, 2024) | 0 | 0 | 0 | 0 | 0 | 0 | 0 | 0 | 0 | 0 |
| | InternVL3-78B (Zhu et al., 2025) | 0 | 0 | 0 | 0 | 0 | 0 | 0 | 0 | 0 | 0 |
| | MiniCPM-o-2.6 (Team, 2025f) | 0 | 0 | 0 | 0 | 0 | 0 | 0 | 2.5 | 0 | 0.2 |
| | kimi-vl-a3b (Team, 2025e) | 0 | 0 | 0 | 0 | 0 | 0 | 0 | 5 | 0 | 0.4 |
| | kimi-vl-a3b-thinking (Team, 2025e) | 0 | 0 | 0 | 0 | 0 | 0 | 0 | 0 | 0 | 0 |

introduction of in-domain exemplars yields negligible improvement. This demonstrates that the key to resolving perceptual limitations does not lie in providing additional knowledge through contextual demonstrations; rather, it points to fundamental architectural deficiencies that cannot be addressed through in-context interventions.

*Table 6.* **In-Context Learning Analysis.** Pass@1 / Pass@32 accuracy (%) with 3-shot ICL. Minimal gains confirm that the bottleneck is representational access, not knowledge acquisition.

| Model | Setting | HiddenText | 3DCaptcha | ColorBlind | ChineseLig. |
|---|---|---|---|---|---|
| Qwen2.5VL-72B | Base | 0 / 0 | 0 / 0 | 0 / 0 | 0 / 0 |
| | +3-shot | 0 / 2.0 | 0 / 0 | 0 / 1.3 | 0 / 0 |
| Gemini 2.5 Pro | Base | 0 / 0 | 0 / 0 | 0 / 0 | 2.5 / 5.0 |
| | +3-shot | 0 / 0 | 0 / 0 | 0 / 4.0 | 7.5 / 20.0 |
| Seed-1.6 | Base | 0 / 0 | 0 / 0 | 0 / 0 | 2.5 / 2.5 |
| | +3-shot | 0 / 0 | 0 / 0 | 0.7 / 0.7 | 0 / 5.0 |

## 5.4. Where Is the Bottleneck? Fine-Tuning Analysis

To isolate the bottleneck location, we conduct SFT on Qwen2.5-7B-VL with five configurations targeting different architectural components. Table 7 reports results.

**Key Finding.** Only configurations updating the vision encoder $f_\theta$ yield substantial gains; updating only the language backbone or adapter produces negligible improvement. This validates Theorem 4.1: *error in text space is proportional to information loss during vision encoding, which cannot be eliminated by enhanced text-space reasoning.* Further details are provided in Appendix E.

**Theoretical Interpretation.** When updating only $\theta$ (freezing $g_\phi, \psi$), the error change approximates:

$$\Delta\varepsilon_{\text{total}} \approx -\frac{1}{2}\Delta\|y - f_\theta(x)\|^2_{\Omega^{-1}} \quad (5)$$

The encoder $f_\theta$ re-parameterizes $\mathcal{X}$ to minimize distance to ground truth $y$, reducing the first term of Eq. 6 while the quantization penalty (second term) remains constant. Improvement correlates directly with enhanced structural fidelity in visual representation:

$$Accuracy \uparrow \iff \mathbb{E}\left[|y - f_\theta(x)|^2\right] \downarrow \quad (6)$$

**Generalization Caveat: Architectural Limitations Beyond Data.** Although fine-tuning the vision encoder greatly improves TET performance, the gains fail to generalize to held-out tasks with different visual styles. This fragility suggests that fine-tuned encoders memorize task-specific patterns rather than learning transferable geometric primitives. Encoder fine-tuning thus locates the bottleneck but does not resolve it—a real solution requires architectures that embed reasoning into perception, maintaining continuous spatial structure rather than collapsing it before reasoning begins.

## 5.5. Whether the Model's Upper Bound Can Overcome the Bottleneck? Reinforcement Learning Analysis

*Table 7.* **Fine-Tuning Component Analysis.** Pass@1 / Pass@32 accuracy (%). Highlighted rows : only updating vision part.

| Parameters Updated | HiddenText | 3DCaptcha | ColorBlind |
|---|---|---|---|
| No Training (Baseline) | 0 / 0 | 0 / 0 | 0 / 1.3 |
| Full Parameters | 90.0 / 94.7 | 95.3 / 98.0 | 77.3 / 79.3 |
| Vision Encoder Only | 86.7 / 94.7 | 94.0 / 98.0 | 87.3 / 98.7 |
| Vision Encoder + Adapter | 82.0 / 94.7 | 95.3 / 97.3 | 99.3 / 99.3 |
| Language Backbone Only | 0 / 2.7 | 0 / 0 | 0.7 / 14.0 |
| Adapter Only | 0 / 5.3 | 0 / 0 | 1.3 / 6.7 |

We use reinforcement learning (RL) as a diagnostic tool to test whether Multimodal Large Language Models (MLLMs) can surpass the previously identified paradigm bottleneck. Experiments are conducted on the OCRVQA dataset (Mishra et al., 2019), using cold-start and GRPO training, where the cold-start data is distilled from Claude-4-Sonnet. We conduct architectural ablations over different optimization scopes—full model, vision tower, and LLM—with and without cold-start alignment, to identify where perceptual reasoning is effectively learned.

**The paradigm bottleneck persists at the performance upper bound.** Consistent with our theoretical analysis (Eq. 6) and prior experimental results (Table. 7), models with reinforcement learning applied solely to the Vision Tower achieve performance on par with full-parameter optimization, while additional optimization of the language

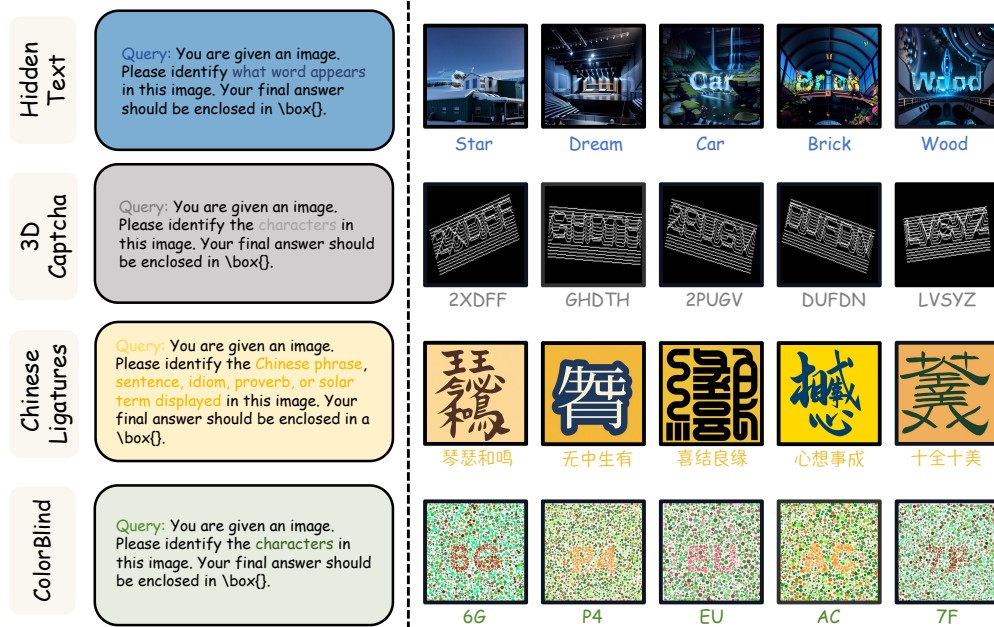

*Figure 2.* Evaluation cases for each TET category: HiddenText, 3DCaptcha, ColorBlind, and ChineseLigatures. The text beneath each image represents the corresponding ground truth. The third line of Chinese characters, read from left to right, symbolizes *marital bliss, a serendipitous union, a perfect match, dreams fulfilled, and flawless.*

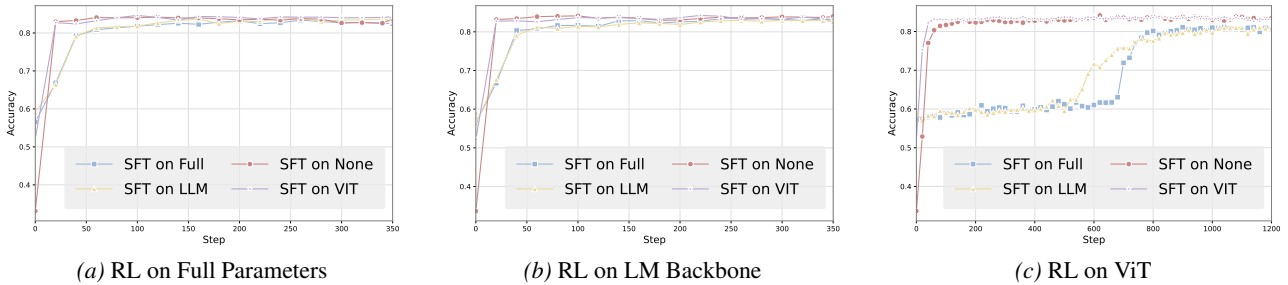

*(a)* RL on Full Parameters      *(b)* RL on LM Backbone      *(c)* RL on ViT

*Figure 3.* **Architecture Ablation on Perception Space.** We conduct architecture ablation experiments during both the cold start and RL stages, where *Full* denotes training the entire architecture, *LLM/ViT* indicates training only the corresponding module, and *None* represents that no cold-start training was performed.

module yields extremely limited gains. This indicates that the performance-limiting factor of the model stems from paradigm design rather than the algorithmic level: optimization on the language side cannot recover the perceptual information lost during the stage of passive visual encoding.

**Upper-bound convergence reveals alignment constraints.** Without cold-start alignment, reinforcement learning rapidly converges to the model's perceptual upper bound; cold-start alignment significantly slows this convergence. This suggests that early alignment biases perceptual tokens toward discrete text space, causing spatial collapse that subsequent reasoning cannot recover.

### 5.6. Resolution Sensitivity: Exposing Semantic Bias

We probe whether failures stem from impaired structural perception or semantic over-reliance by evaluating HiddenText under resolution perturbations (Fig. 5): (1) direct downsampling, which removes fine-grained detail while preserving coarse semantics, and (2) downsampling then upsampling, which preserves global layout but disrupts local geometry.

Under direct downsampling, accuracy rises as resolution drops—revealing reliance on coarse semantics over fine structure. Yet structural blurring hurts performance despite intact global layout, exposing dependence on superficial texture cues rather than true geometry. Both patterns confirm semantic bias: spatial details critical to the task are collapsed in text-aligned representations and irrecoverable by later reasoning.

## 6. Toward Reasoning Within Perception

Our diagnosis points to a clear prescription: **visual reasoning must occur in visual space, not text space.** Rather

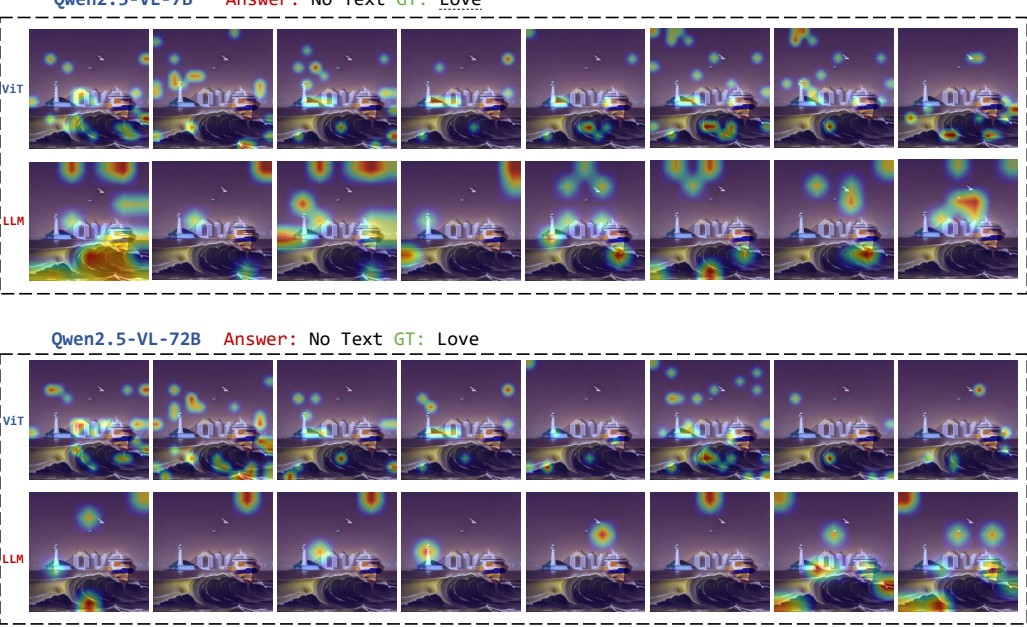

*Figure 4.* Grad-CAM of Qwen2.5-VL Series Models on HiddenText .

than advocating for a single architecture, we outline the design principles that any solution must satisfy to escape the information collapse bound.

### 6.1. Design Principles for Native Visual Reasoning

We identify 3 core principles distinguish *reasoning within perception* from prevailing *reasoning about perception*:

**Principle 1: Reason Before Collapse.** Reasoning must occur before visual information is projected into text-aligned embeddings. Once compressed into text space, pixel-level details and geometric relationships are irreversibly lost—reasoning on post-compression representations is already too late.

**Principle 2: Persistent Visual Access.** The original visual representation must remain accessible throughout the reasoning process, allowing repeated queries to the visual substrate rather than a one-shot encoding that is subsequently discarded.

**Principle 3: Reasoning-Driven Perception.** High-level semantic hypotheses should actively guide which visual regions or relationships to examine. Perception should be steered by reasoning demands, not merely consumed as a fixed, pre-computed input.

### 6.2. A Formal Framework: Active Visual Querying

To formalize how these principles escape the information collapse bound, we propose **Active Visual Querying (AVQ)**—a mechanism where reasoning can issue queries back to the visual representation, and where perception itself is steered by reasoning demands.

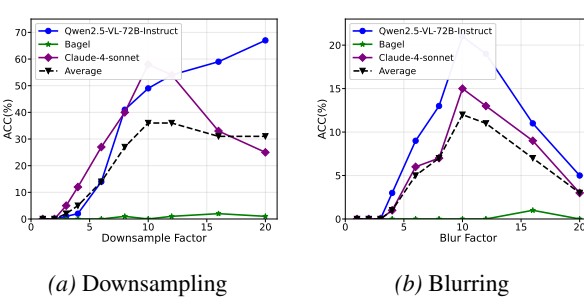

*(a)* Downsampling      *(b)* Blurring

*Figure 5.* **Resolution Sensitivity.** (a) Downsampling paradoxically improves accuracy, suggesting reliance on coarse semantics over fine structure. (b) Blurring degrades performance despite preserved global layout, confirming dependence on local texture cues.

**Mechanism Formulation.** Let $z = f_\theta(x)$ denote the visual representation and $s_t$ the reasoning state at step $t$. AVQ operates as:

$$a_t = \mathcal{A}(z, \mathcal{Q}(s_t); x), \quad s_{t+1} = \mathcal{U}(s_t, a_t) \quad (7)$$

Here $\mathcal{Q}$ generates a visual query from the current reasoning state at each step, and $\mathcal{A}$ retrieves task-relevant information from $z$ (e.g., re-attending to specific regions, enhancing task-relevant features). $\mathcal{U}$ then updates the reasoning state. The final output is $\hat{y} = h_\psi(s_T)$.

The key requirements are: (1) $\mathcal{A}$ operates in visual space $\mathcal{Z}$ rather than text space $\mathcal{E}$, and (2) retrieval is conditioned on the query $\mathcal{Q}(s_t)$, making perception an active, reasoning-driven process rather than fixed encoding. This implements all three principles: reasoning occurs before collapse (Prin-

ciple 1), visual access persists throughout (Principle 2), and perception is steered by reasoning demands (Principle 3).

**Architectural Gap.** Current MLLMs partially approximate AVQ: cross-attention to image tokens provides a form of retrieval $\mathcal{A}$, but both query generation $\mathcal{Q}$ and perception refinement $\mathcal{P}$ are implicit and fixed per layer—not explicitly driven by the reasoning process. Tool-augmented approaches implement explicit $\mathcal{Q}$ but through discrete API calls that sacrifice differentiability. Crucially, no current architecture allows reasoning to *steer perception itself*—the visual encoding $z$ remains static regardless of downstream demands. Closing this gap is a concrete research direction.

### 6.3. Architectural Instantiations

The AVQ framework admits multiple architectural realizations—cross-attention to pixels, recurrent visual transformers (Zhang et al., 2025a), and generative refinement (Chung et al., 2025)—each implementing partial aspects of the query-retrieval-update cycle. However, no existing architecture fully instantiates AVQ with explicit query generation, reasoning-driven retrieval, and grounded state update. We detail these instantiations in Appendix H.

### 6.4. What This Position Does Not Claim

We emphasize that our position does not require abandoning language as an output modality or reasoning medium. Many visual tasks *are* well-served by text-space reasoning: captioning, VQA on semantic content, or problems where linguistic structure provides useful inductive bias. For such tasks, the standard pipeline—encoding once, then reasoning in text space—remains appropriate.

Our claim is narrower: for tasks requiring **fine-grained spatial grounding**—geometry, topology, precise counting, occlusion reasoning—the architecture must support reasoning in visual space *when needed*. The goal is not to replace vision-language integration, but to provide flexibility when task-critical spatial information would otherwise be lost.

## 7. Alternative Views

We acknowledge and address several perspectives that challenge our position:

**"Scale Will Solve It."** Our benchmark results (Table 5) provide direct evidence against this view: performance remains near-zero across model scales from Qwen2.5VL-7B to 72B. Even with RLVR-scaled reasoning, SOTA models fail on fundamental visual primitives (Chen et al., 2026; Wang et al., 2026b; Chen et al., 2025a; Li et al., 2025c)—the bottleneck is information-theoretic, not computational (Liu et al., 2026; Weng et al., 2025).

**"Better Vision Encoders Suffice."** Our evidence partially

supports this—fine-tuning the encoder helps substantially. However, these gains often fail to generalize beyond the training distribution (Zhai et al., 2023; Wu et al., 2025b; Wang et al., 2025d), addressing only "what to see," not "when to reason." A better encoder still faces the sequential bottleneck if reasoning remains deferred to text space (Wang et al., 2026a).

**"Tool Use Is the Answer."** Tool-augmented approaches (Surís et al., 2023; Lai et al., 2025) extend capabilities, but perception is accessed only via discrete API queries and structured outputs, not a continuously manipulable visual state (Ke et al., 2025; Li et al., 2025b). This is augmentation, not native visual reasoning.

More alternative views can be found in Appendix F.

## 8. Implications and Future Directions

Our position suggests several directions for future research, including architectural designs that prioritize reasoning within visual representations, training objectives that preserve spatial fidelity beyond semantic alignment, benchmarks that isolate perceptual from reasoning failures, and adaptive routing between visual and text pathways based on task demands. These directions share a common principle: enabling computation to *return* to visual space when task demands require it. We elaborate on these in Appendix G.

## 9. Conclusion

In this paper, we have argued that persistent visual failures in MLLMs stem from a structural misalignment: deferring reasoning to language generation displaces computation from the visual manifold to textual space, collapsing task-critical spatial signals before deliberation begins. Our theoretical analysis bounds this information loss, and the Turing Eye Test confirms that vision encoder modifications produce dramatic improvements while inference-time interventions yield minimal gains—implicating the perceptual bottleneck rather than reasoning capacity. These findings motivate a paradigm shift from reasoning about perception to reasoning within perception: architectures must enable deliberation on pixel-level representations before semantic compression, not after. Only then can we bridge the gap between fluent visual description and faithful visual understanding.

### Acknowledgements

This work is supported by Fundamental and Interdisciplinary Disciplines Breakthrough Plan of the Ministry of Education of China (JYB2025XDXM113), National Natural Science Foundation of China (92470121, 62402016), National Key R&D Program of China (2024YFA1014003), Zhongguancun Academy (C20250204, C20250602), and Beijing Major Science and Technology Project (Z251100008125043, Z251100008425023).

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

# A. TET Dataset Generation and Evaluation Details

## A.1. Generation Details

**HiddenText.** We render text characters using composite image elements that appear as recognizable letters/words when viewed at reduced scale but resolve into detailed scenes when magnified. The generation pipeline involves: (1) selecting target text strings, (2) decomposing each character into constituent visual elements, (3) replacing elements with thematically consistent image patches, and (4) compositing at multiple scales to verify the dual-interpretation property.

**3DCaptcha.** Characters are rendered with 3D perspective distortion and curved surfaces. We apply: (1) random rotation around all three axes, (2) perspective projection with varying focal lengths, (3) surface curvature following Bezier curves, and (4) realistic shading and shadow effects.

**ColorBlind.** Building on Ishihara plate design (Ishihara, 1951), we: (1) define target characters, (2) generate foreground dots with target-distinguishing chromaticity, (3) add background dots with confounding colors at similar luminance, and (4) introduce additional noise dots that share partial chromatic similarity with the target.

**ChineseLigatures.** We synthesize fused glyphs by: (1) decomposing source characters into radicals and strokes, (2) applying morphological transformations (scaling, rotation, skewing), (3) spatially compositing multiple characters with overlap, and (4) applying calligraphic style transfer for visual coherence.

## A.2. Evaluation Details

We evaluate 15 state-of-the-art models spanning three categories: (1) **Unified multimodal models**: Show-o2 (Xie et al., 2025), Bagel (Deng et al., 2025), Janus-Pro (Chen et al., 2025b); (2) **Closed-source APIs**: Claude 4-Sonnet (Anthropic, 2025), Gemini 2.5 Pro (Team, 2025b), OpenAI o1 (Jaech et al., 2024), Seed-1.6 (Team, 2025a); (3) **Open-source models**: Qwen2.5VL-72B (Bai et al., 2025b), QVQ-72B (Team, 2024), InternVL3-78B (Zhu et al., 2025), MiniCPM-o-2.6 (Team, 2025f), kimi-vl (Team, 2025e). We maintain original inference settings for unified models and configure all others with temperature 0.3 and 16384 max tokens.

# B. Detailed Pass@K Metrics for TET and More Response Case

Fig. 6 reports the pass@K curves across tasks, together with their standard deviations. As K increases, existing MLLMs show consistently flat performance trajectories: average accuracy exhibits little variation across tasks, and even the best-performing cases achieve gains of less than 4%.

This behavior indicates that these tasks are not amenable to improvement through expanded exploration in the *reasoning space*, in contrast to prior benchmarks where higher K often yields substantial gains. The uniformly flat curves suggest that the dominant bottleneck lies not in insufficient reasoning diversity or search depth, but in the models' inability to access and preserve task-critical visual evidence. The failure cases in Fig. 7 further substantiate this diagnosis, with additional examples provided in Fig. 15.

# C. Interpretation of MLLMs with Grad-CAM

To investigate why models, despite possessing sophisticated linguistic reasoning capabilities, still fail to perceive images accurately, we conducted Grad-CAM analyses on all datasets following the protocols of prior work (Selvaraju et al., 2017; Zhang et al., 2024a). We systematically examined two repre-

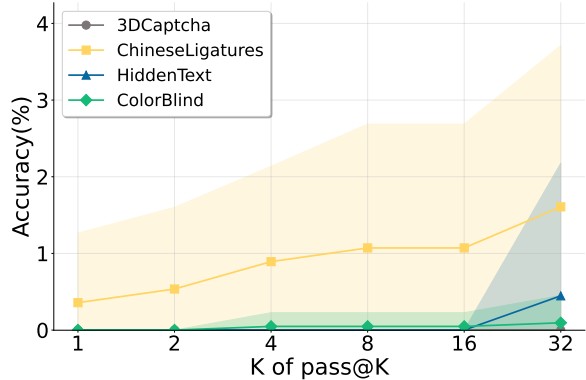

*Figure 6.* **Pass@k Results.** Mean and variance on four TET tasks.

sentative models from the Qwen2.5-VL series (with 7B and 72B parameters), analyzing the attention maps of both the visual backbone network and the language decoder components. Attention patterns were uniformly sampled across multiple layers to capture the evolutionary process of attention throughout the architectural depth of each component. This analysis revealed a key insight: as visual reasoning is deferred to the language decoder downstream of the visual encoding bottleneck,

the spatial focus of attention becomes misaligned with the task-critical visual evidence. Consequently, no matter how the parameter count of the language decoder is scaled up, the model still cannot perceive images accurately.

**Information flow in the visual encoder.** As illustrated in Figures 4, 8, 9, and 10, the visual encoder operates without access to downstream reasoning requirements, functioning solely to extract generic image representations for the LLM decoder. This embodies the passive, one-off feature encoding process critiqued in our framework: the encoder must compress visual information into a fixed representation before knowing what spatial distinctions the reasoning stage will require. This temporal-spatial mismatch manifests in systematic attention failures. While the ViT allocates attention across various image regions, this attention frequently falls outside target character regions or captures only partial segments, prioritizing salient object-level features over task-critical spatial details. Because the decision of when to reason (deferred to language generation) is made before knowing where visual evidence resides, the encoder cannot preserve the fine-grained geometric and topological distinctions that these tasks require. In the *3DCaptcha* task, the ViT exhibits region-specific rather than global recognition patterns, demonstrating how this passive encoding paradigm degenerates into a lossy compression that discards task-critical spatial signals before reasoning begins—signals that cannot be recovered through subsequent language-side deliberation.

**Information flow in the LLM decoder.** As shown in Figures 4, 8, 9, and 10, the LLM decoder—where actual reasoning occurs in current architectures—exhibits consistent attention failures across both model scales (7B and 72B). Except for the *ChineseLigature* task, the decoder systematically fails to attend to regions containing task-critical visual information, instead scattering attention over irrelevant areas or entirely missing essential visual elements. This pattern reveals the core structural problem: by the time reasoning begins in the decoder, the visual representation has already been collapsed into a text-aligned interface that no longer preserves the pixel-level distinctions necessary for correct decisions. The decoder cannot "look back" to recover lost spatial information; it can only reason within the impoverished visual representation it receives. What is needed is not passive encoding followed by language reasoning, but rather active, reasoning-driven perception where the encoding process itself is guided by task-specific requirements—allowing high-level reasoning to dynamically query and steer visual feature extraction to preserve the precise spatial evidence needed for decision-making.

**Speculations on the causes of failure across different tasks.** These failures demonstrate that task-critical spatial signals are collapsed before reasoning begins and cannot be recovered through language-side deliberation.

As shown in Fig. 4, for *HiddenText*, MLLMs fail to recognize symbols formed by spatially distributed objects, indicating that geometric relationships and global compositional structure are not preserved through the text-aligned encoding bottleneck. The passive encoding captures local features but discards the relational topology needed for integration.

As shown in Fig. 8, for *3DCaptcha*, models cannot disambiguate overlapping characters, demonstrating that depth cues, occlusion boundaries, and figure-ground segmentation—measurement-like operations requiring continuous spatial reasoning—cannot be executed in discrete textual space.

As shown in Fig. 9, for *ColorBlind*, MLLMs struggle to construct characters from same-colored dots scattered among

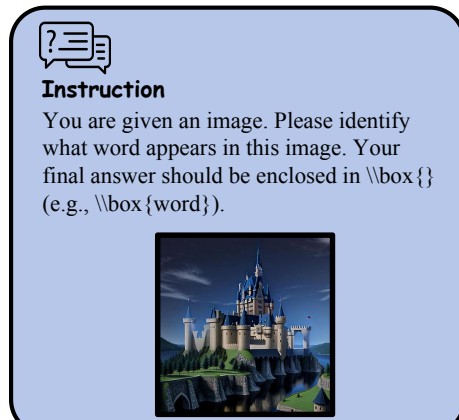

**Figure 7. Model response on question of HiddenText.** The goal is to identify the hidden word in an image. Gemini-2.5-Pro-0506 answers the hidden word as "castle".

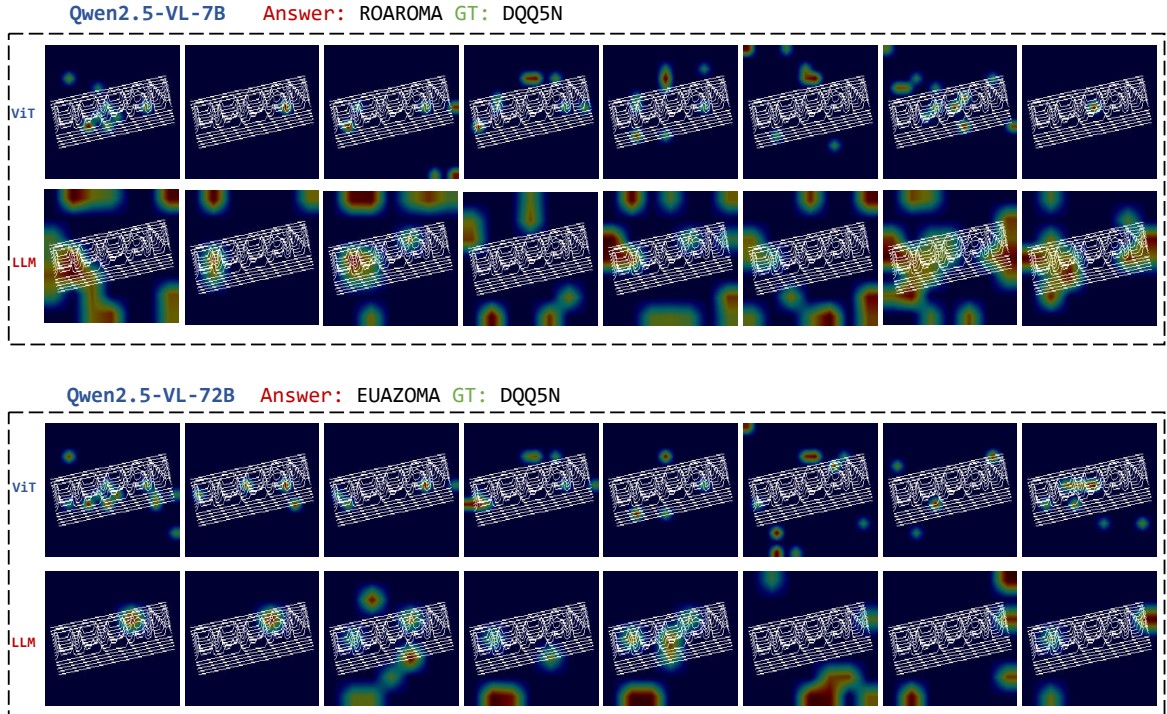

*Figure 8.* Grad-CAM of Qwen2.5-VL Series Models on 3DCaptcha.

noise, illustrating that perceptual grouping—conceptually simple but hard to verbalize—requires active visual reasoning on pixel-level representations.

As shown in Fig. 10, for *ChineseLigature*, MLLMs perceive individual components but cannot compositionally extend them into plausible phrases, revealing that fine-grained spatial arrangements—distinctions hard to specify in text—are lost during one-off encoding.

## D. Grad-CAM Analysis of Fine-Tuned MLLMs

Fig. 11, 12, 13 presents Grad-CAM visualizations of Qwen2.5-VL-7B before and after vision module fine-tuning across different datasets. Following vision module fine-tuning, the model demonstrates enhanced perceptual capabilities, as attention coverage over effective character regions across inter-module interactions increases. This phenomenon validates that targeted optimization of the vision module effectively improves the generalization of the model's perceptual patterns.

## E. Further details of supervised fine-tuning

To validate that visual perception, rather than language reasoning, is the primary bottleneck, we incorporate traditional benchmarks—OCR-VQA, GeoQA, and CLEVR—for comparison. As shown in Fig. 14, on TET tasks, configurations excluding visual fine-tuning plateau early, while those including it converge efficiently. In contrast, all configurations converge similarly on traditional benchmarks. This divergence reveals that traditional tasks fall within current MLLM pre-training coverage, requiring only language-side improvements, while TET demands enhanced visual perception that cannot be compensated by scaling language reasoning alone. This supports our claim that the limiting factor is representational access—visual fine-tuning preserves task-critical spatial distinctions that passive encoding collapses before reasoning begins.

## F. More Alternative Views and Counterarguments

**"Perception and Language Should Remain Separate."** From a modularity perspective, clean separation aids interpretability and compositionality. We acknowledge this concern but note that biological vision does not maintain such separation (Marr, 1982). The question is whether the interface between modules preserves sufficient information—our evidence suggests

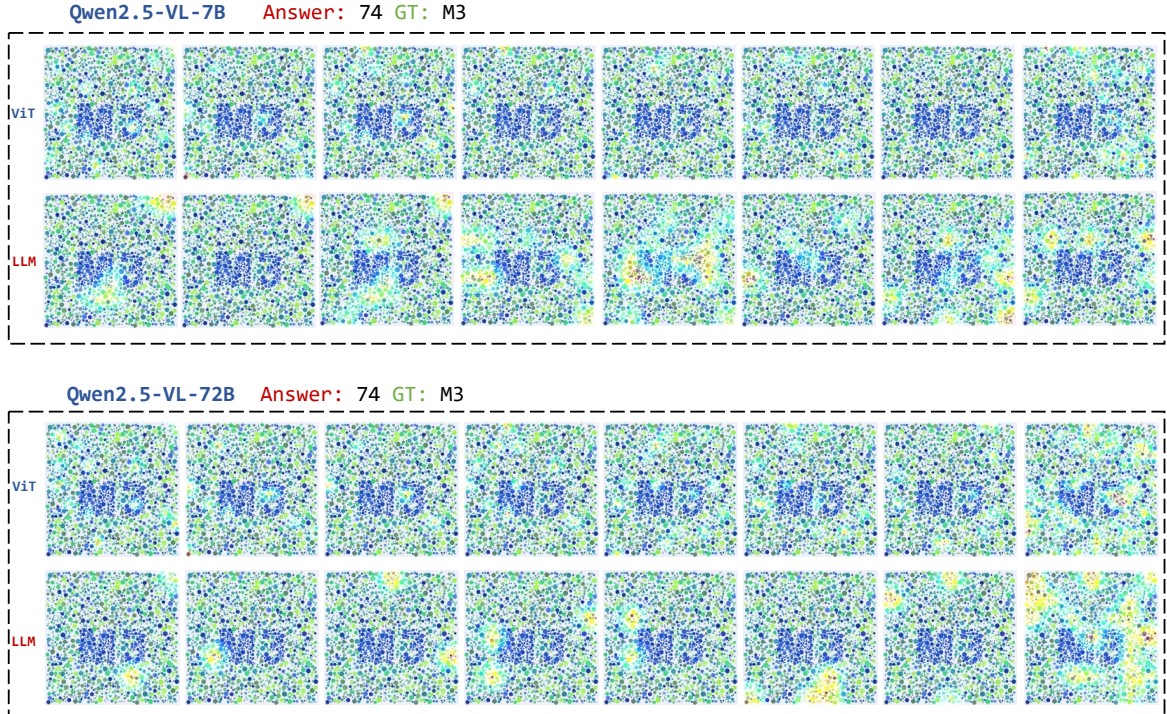

*Figure 9.* Grad-CAM of Qwen2.5-VL Series Models on ColorBlind subset.

current interfaces do not for fine-grained visual tasks. (Cartuyvels et al., 2021; Nagrani et al., 2021)

**"These Tasks Are Artificial."** One might dismiss TET tasks as contrived edge cases. We counter that these are *diagnostic*: they isolate specific failure modes that manifest subtly in realistic settings. Failures on 3DCaptcha predict failures on real-world 3D reasoning; failures on ChineseLigatures predict failures on fine-grained document understanding (Mathew et al., 2021). TET is a stress test, not an end goal.

## G. Details of Implications and Future Direction

Our position suggests several directions for future research:

**Architectural Design.** The design space should prioritize mechanisms for reasoning within visual representations—recurrent refinement, cross-attention back to pixels, or hybrid continuous-discrete representations—over scaling text-side computation.

**Training Objectives.** Contrastive objectives (e.g., CLIP) optimize for semantic alignment at the expense of structural fidelity. Future objectives should explicitly preserve fine-grained spatial information required for downstream reasoning, potentially through multi-scale reconstruction losses or geometric consistency constraints.

**Benchmark Development.** Current benchmarks often conflate perceptual and reasoning capabilities. We advocate for diagnostic probes like TET that isolate perceptual failures, enabling precise identification of architectural bottlenecks.

**Adaptive Routing.** Not all tasks require pixel-level reasoning. Future architectures might learn to route computation through visual vs. text pathways based on task demands—reasoning in visual space for spatial tasks, in text space for semantic tasks.

## H. Architectural Instantiations of AVQ

The AVQ framework admits multiple concrete realizations, each implementing the query-retrieval-update cycle $(\mathcal{Q}, \mathcal{A}, \mathcal{U})$ with distinct trade-offs:

**Cross-Attention to Pixels.** The reasoning module repeatedly attends back to the original image (or early feature maps) when resolving ambiguous predicates. Here $\mathcal{Q}$ is implicit in the query vectors of attention layers, $\mathcal{A}$ is cross-attention over

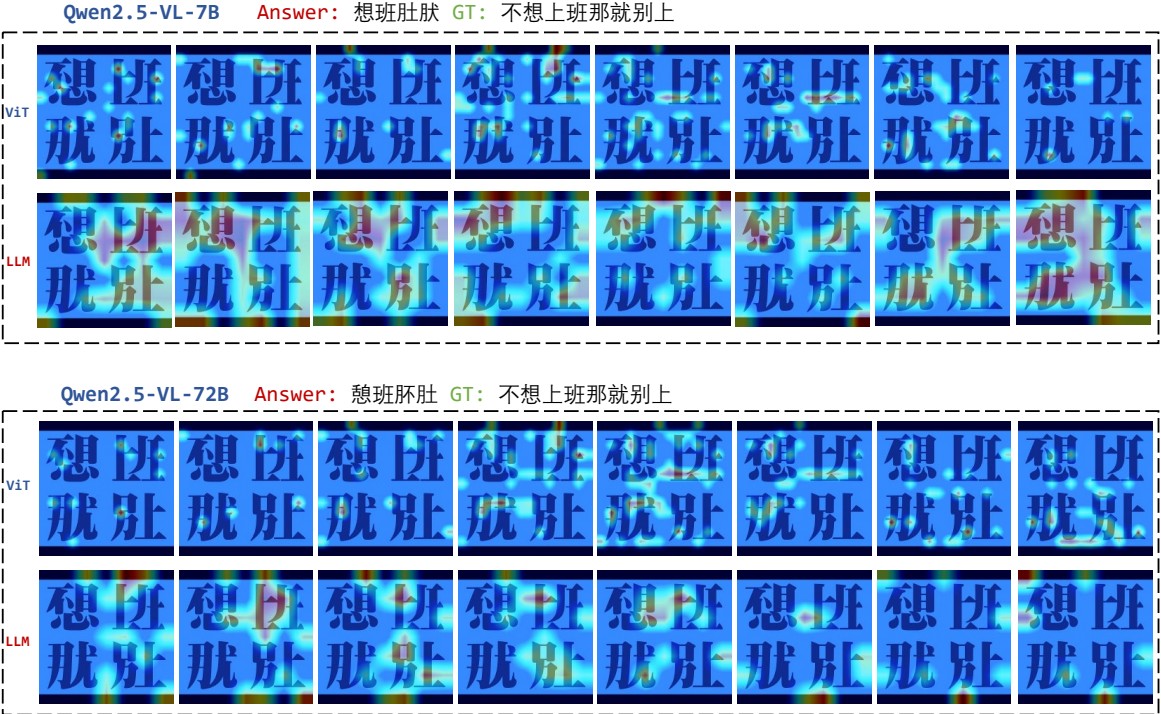

*Figure 10.* Grad-CAM of Qwen2.5-VL Series Models on ChineseLigature subset.

visual tokens, and $\mathcal{U}$ is the residual update. This maintains pixel-level access but lacks explicit query generation ($\mathcal{Q}$) and perception refinement ($\mathcal{P}$)—the visual representation remains static.

**Recurrent Visual Transformers.** Stacking transformer blocks with shared weights enables iterative refinement within visual latent space (Zhang et al., 2025a). The recurrence implements $\mathcal{P}$ and $\mathcal{U}$ through repeated processing, but $\mathcal{Q}$ remains implicit—refinement is not explicitly conditioned on semantic hypotheses from the reasoning process.

**Generative Refinement.** Diffusion-based models that "re-render" interpretations in visual space (Chung et al., 2025) implement $\mathcal{P}$ and $\mathcal{U}$ as iterative denoising, maintaining pixel-level fidelity throughout. However, the conditioning signal acts as a fixed prompt rather than dynamically generated queries—$\mathcal{Q}$ is absent.

No existing architecture fully instantiates AVQ with explicit query generation ($\mathcal{Q}$), reasoning-driven retrieval with perception refinement ($\mathcal{A}$), and grounded state update ($\mathcal{U}$). This gap defines a concrete research agenda.

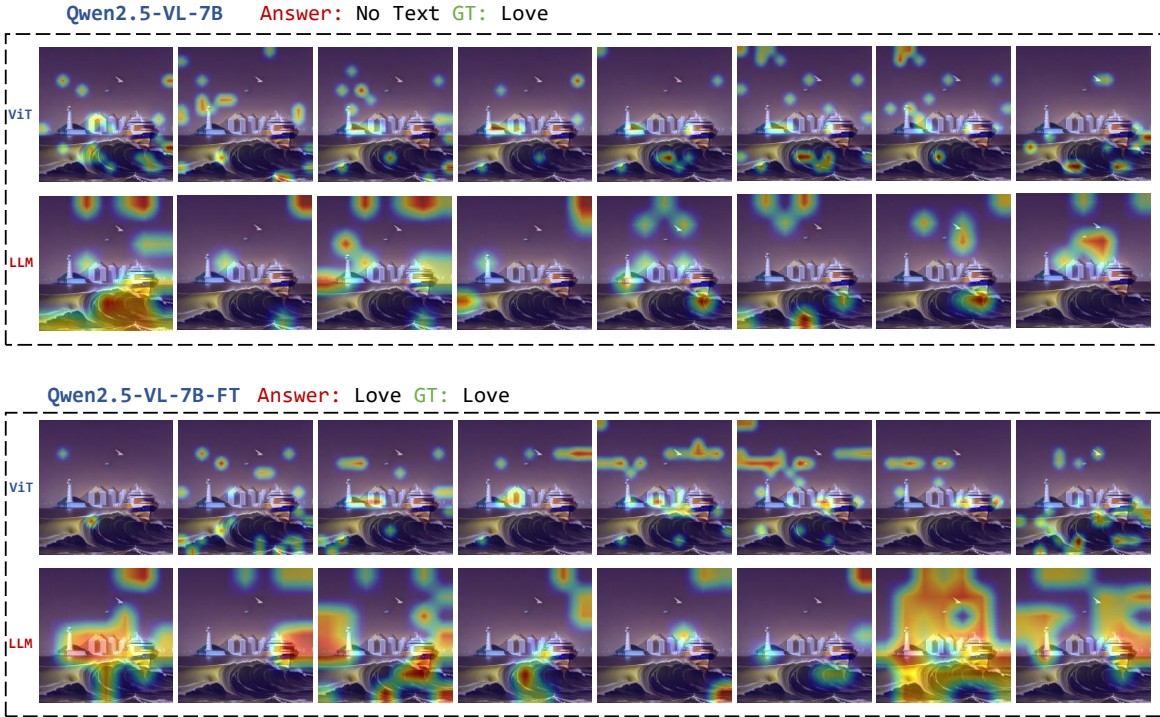

*Figure 11.* Grad-CAM of Qwen2.5-VL-7B before and after visual fine-tuning on HiddenText.

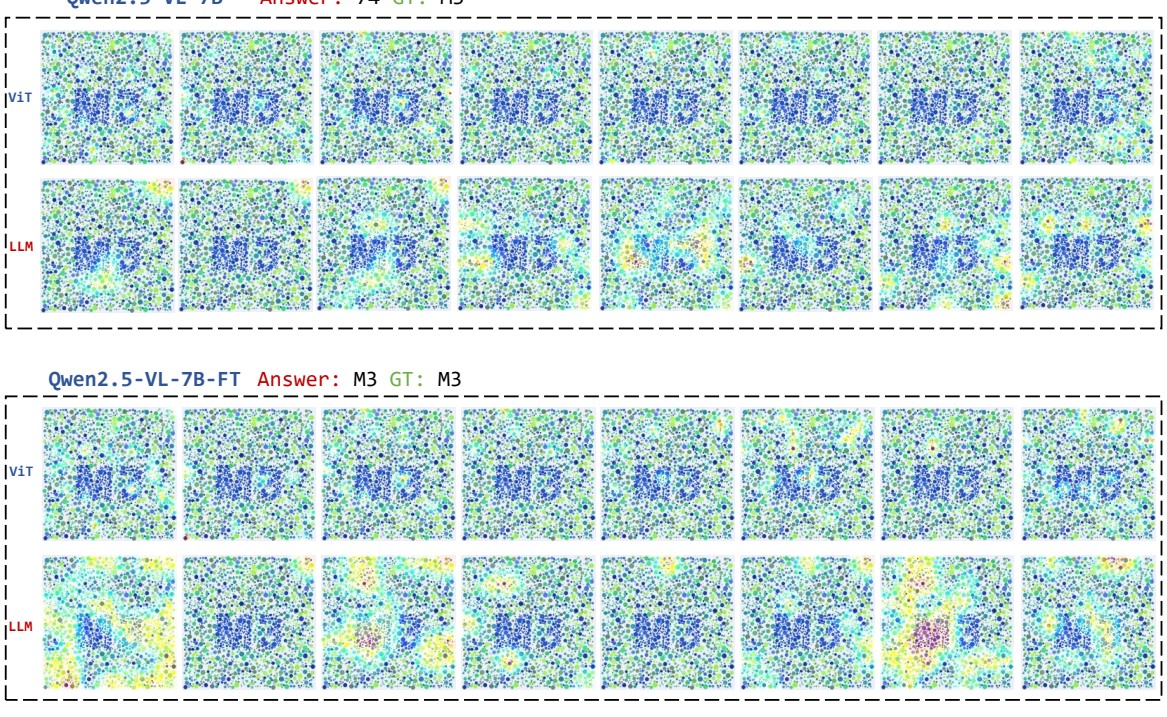

*Figure 12.* Grad-CAM of Qwen2.5-VL-7B before and after visual fine-tuning on ColorBlind.

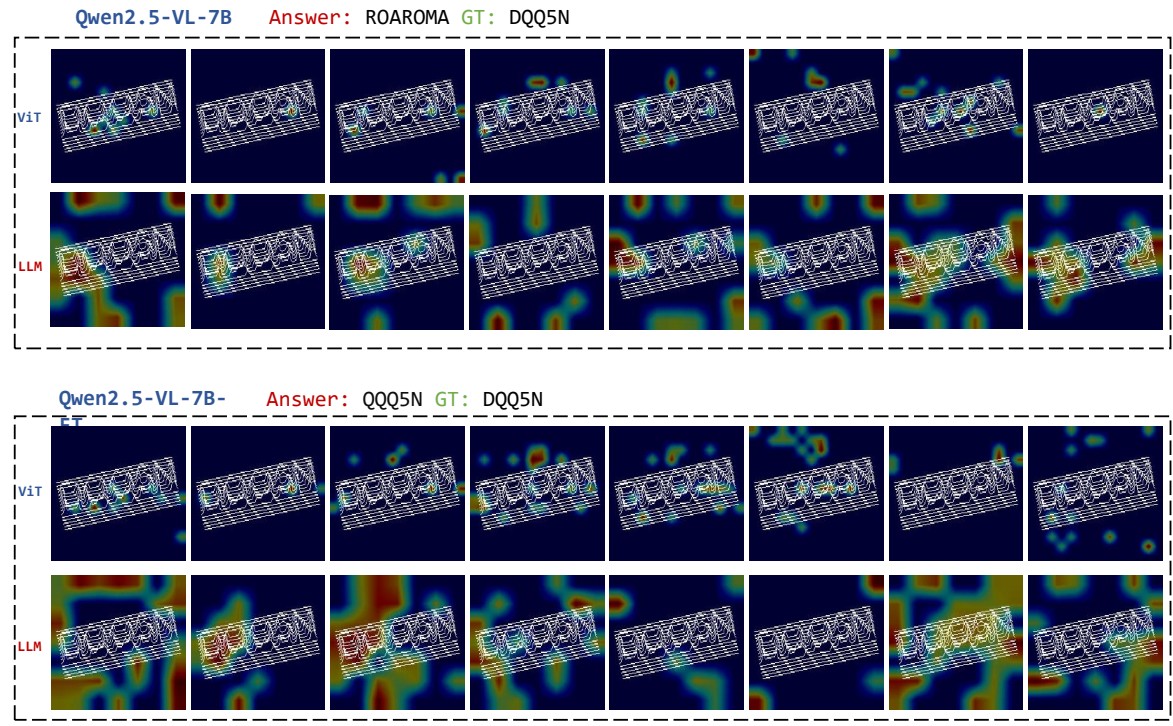

*Figure 13.* Grad-CAM of Qwen2.5-VL-7B before and after visual fine-tuning on 3DCaptcha.

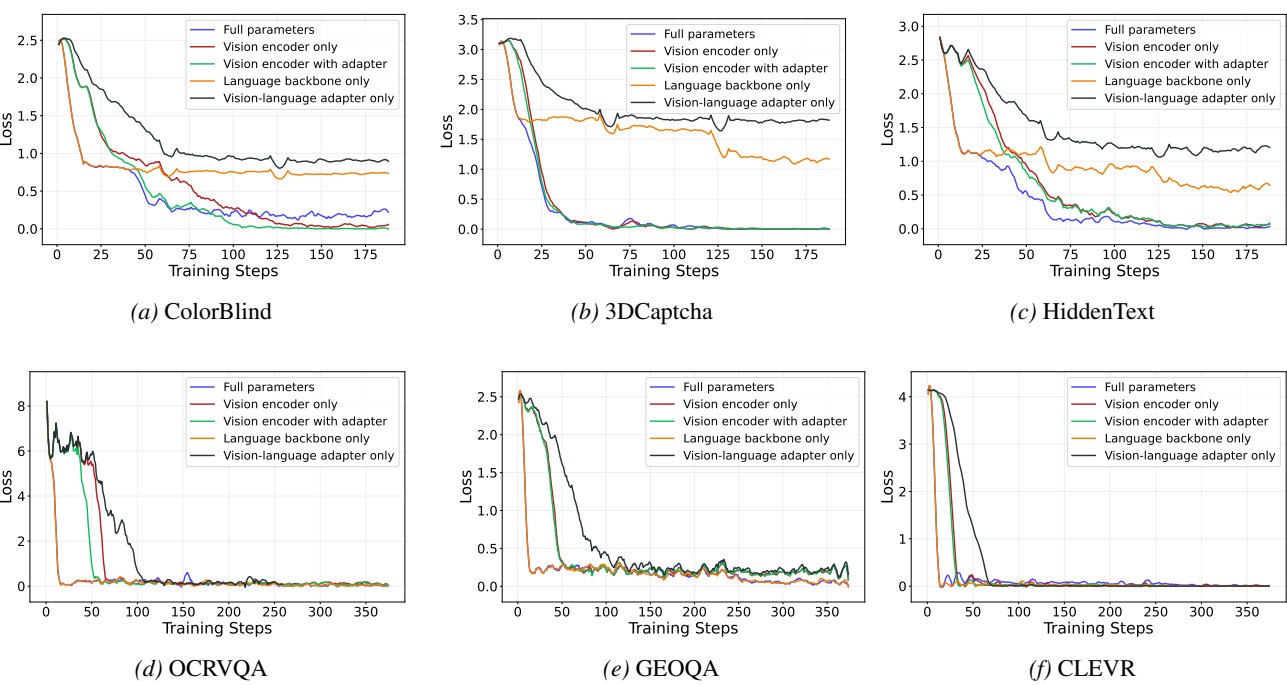

*(a)* ColorBlind  *(b)* 3DCaptcha  *(c)* HiddenText

*(d)* OCRVQA  *(e)* GEOQA  *(f)* CLEVR

*Figure 14.* **Training Curves.** Training loss curve for different settings on finetuning parameters for both our tasks and traditional tasks.

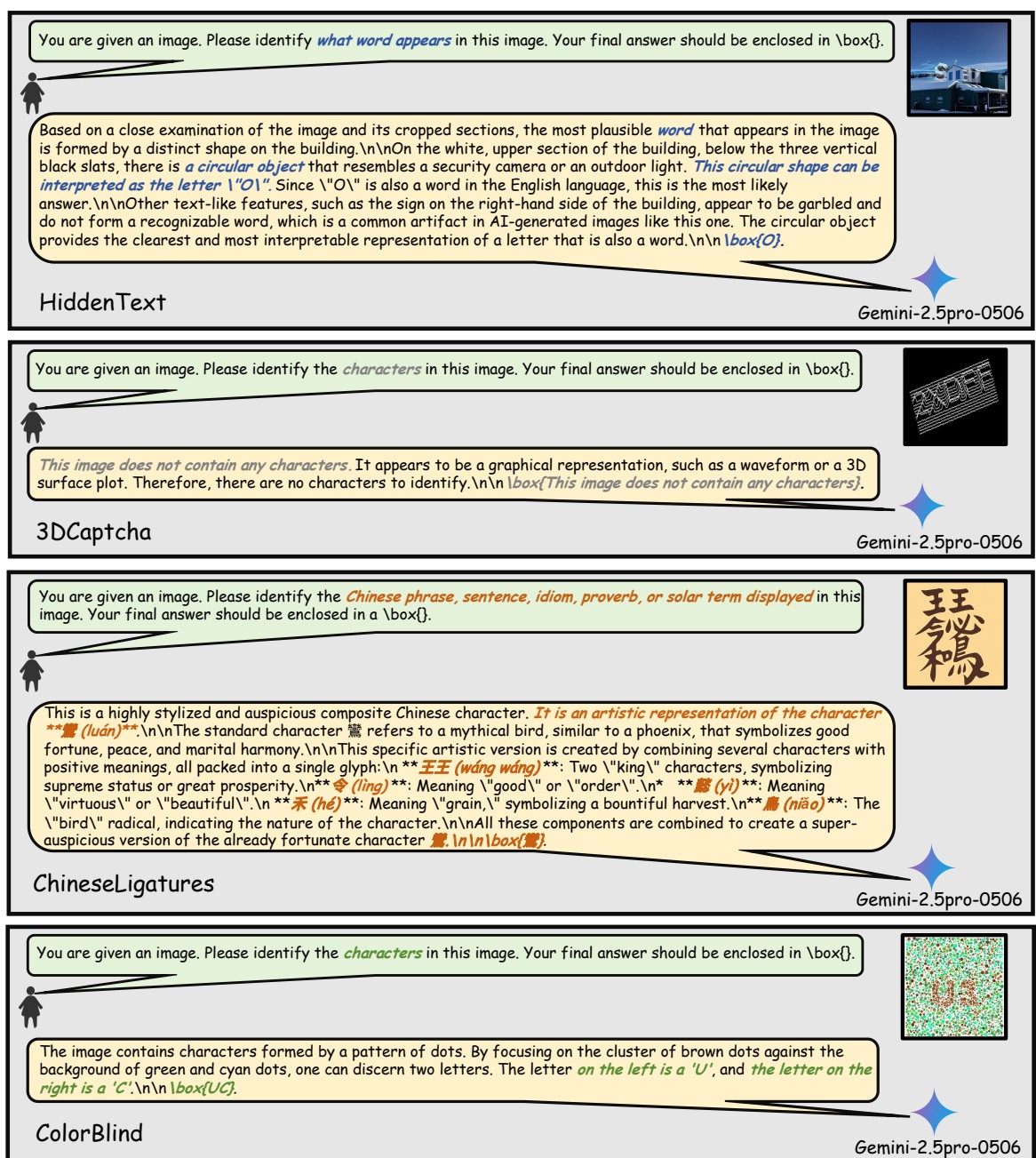

*Figure 15.* Example responses from the Gemini model on the four tasks of TET. For each task, the model fails to provide correct analysis and conclusion based on its flawed initial perception.

