# OpenReview forum: "Position: Reasoning After Perception Means Reasoning Without Vision"
_ICML.cc/2026/Position_Paper_Track — ICML 2026 Position Paper Track regular_

### Official Review · Reviewer_HvMT · 2026-03-12

**Significance:** 3
**Argument Clarity:** 3
**Rating:** 5
**Confidence:** 4

**Questions:**

See above

**Alternative Views Section:**

Yes

**Compliance With Llm Reviewing Policy A Conservative:**

Affirmed.

**Discussion Potential:**

3

**Final Justification:**

I recommend accept with most concerns resovled.

**Paper Summary:**

The paper challenges the prevailing "Perception-then-Reasoning" paradigm in Multimodal Large Language Models (MLLMs), arguing that deferring visual reasoning to the language decoder causes a structural "Information Collapse". The authors contend that current architectures displace computation from the continuous visual manifold to a discrete textual space, effectively rendering them "shape-blind" for tasks that are hard to verbalize but visually simple. To substantiate this, they introduce the Turing Eye Test (TET), a diagnostic benchmark targeting fine-grained perception (e.g., 3D distortions, Ishihara-style color tests, and scale-variant text). Their findings show that scaling language-side deliberation (like Chain-of-Thought) fails to recover lost spatial signals. Consequently, they advocate for "Reasoning Within Perception," an architectural shift that allows models to actively query and steer pixel-level representations before semantic compression occurs

**Position:**

Yes

**Position In Title:**

Yes

**Related Work:**

3

**Strengths And Weaknesses:**

Strengths:
1. Strong Theoretical Grounding: The paper formalizes the "Information Collapse" through an information-theoretic bound, proving that the projection from visual features to text-aligned latent space is semantically contractive and non-invertible for geometric details.
2. Diagnostic Benchmark: The introduction of the Turing Eye Test (TET) provides a rigorous way to isolate perceptual failures from high-level semantic reasoning, exposing "collective blindness" even in state-of-the-art models like Gemini 2.5 Pro and GPT-o1.
3. Rigorous Empirical Validation: The authors conduct comprehensive architectural ablations, showing that only updates to the vision encoder, rather than the language backbone or adapter, yield substantial gains on perceptual tasks.
4. Clear Architectural Prescription: Instead of just identifying a problem, the paper proposes a formal framework called Active Visual Querying (AVQ), which outlines three core principles for future native visual reasoning: reasoning before collapse, persistent visual access, and reasoning-driven perception

Weankess:
1. The specially designed TET test is mostly composed of images with details hard to recognize (e.g., images for color blindness or images containing infrequent characters). The difficulty may be largely dependent on the professionality and scarcity of corresponding required knowledge for the vision encoder to finish this task, instead of acquiring vision details. Experiments also show training the vision encoder on these images could notably improve the model performance, which equips the vision encoder with corresponding knowledge. Further experiments or analysis should be given to show it's actually the lack of information contained in the input images causes the model to performance worse.

2. Because tool-augmented models exbihit as strong competitors for the idea proposed by this paper, which can also use external tools to grab raw information from input images, the authors should incorporate these models for comparison in the main results, i.e., tab.5.

3. Some tables or figures are just placed without correct citations in the main text, e.g. tab.2 and fig.4, which make the reader confused with no explanations.

4. The paper defines the requirements for a solution (AVQ) and critiques current approximations (like cross-attention to pixels), but it does not present a novel architecture overview or insights that try to sove the "Information Collapse" bound. It's better to more thoroughly describe the final proposed framework, i.e., Active Visual Querying, with a framework overview or more texts.

**Support:**

3

---

> ### Author Rebuttal · Authors · 2026-03-31
>
> Thank you for your supportive review and suggestions. Below we respond to the comments in **Weaknesses (W)**.
>
> ---
>
> **W1. Does TET Difficulty Stem from Knowledge Gaps Rather Than Visual Information Loss?**
>
> We agree that some TET examples require prior knowledge, meaning this factor cannot be entirely excluded. However, our experiments support a critical conclusion: **even when knowledge factors are present, the retention and access of visual evidence constitute an independent and dominant performance bottleneck**.
>
> Specifically, Table 6 shows that 3-shot ICL barely recovers performance, indicating that supplementing text examples is insufficient. Furthermore, Table 7 demonstrates that only vision-side updates bring significant gains, while language-only and adapter-only updates yield limited effects. This isolates the bottleneck to **visual access pathways** rather than language knowledge injection. Combined with the attention analysis in the Appendix, these results confirm that current failures cannot be fully explained by knowledge deficits. Limitations in visual representation and access remain a necessary explanation.
>
> ---
>
> **W2. Tool-Augmented Baselines in Table 5**
>
> We agree that tool-augmented models are a relevant comparison, as they likewise suggest that some multimodal failures require reasoning over image evidence rather than only text-aligned representations. We therefore evaluated OpenAI o3 on TET and will include it in the revised Table 5. Its performance remains low overall (Avg. Pass@1 = 0.82%, Avg. Pass@32 = 4.49%), suggesting that **tool access alone does not resolve the fine-grained perceptual failures** targeted by TET. While specialized tools may help, current systems often decide whether to call a tool, which tool to call, and what to query from already compressed image representations, so tool use itself may be misguided. We thus view tool augmentation as **complementary rather than contradictory**: it re-accesses visual evidence externally, whereas our framework seeks to keep reasoning grounded in the original image throughout inference.
>
> | Model | HiddenText Pass@1 | HiddenText Pass@32 | 3DCaptcha Pass@1 | 3DCaptcha Pass@32 | ColorBlind Pass@1 | ColorBlind Pass@32 | ChineseLigatures Pass@1 | ChineseLigatures Pass@32 | Average Pass@1 | Average Pass@32 |
> |-------|:-----------------:|:------------------:|:----------------:|:-----------------:|:-----------------:|:-----------------:|:-----------------------:|:------------------------:|:--------------:|:---------------:|
> | OpenAI o3 | 0% | 0.67% | 2.67% | 12% | 0% | 0% | 0% | 7.50% | 0.82% | 4.49% |
> ---
>
> **W3. Unclear In-text Citations for Table 2 and Fig. 4**
>
> We appreciate the reviewer for identifying this oversight. We would like to clarify that Fig. 4 has been cited and discussed in Sec. 5.6, and we will present it more clearly in the revision. We apologize for missing the citation to Table 2, and we will add it near the end of Sec. 2, where we summarize the benchmark landscape before the structural critique.
>
> ---
>
> **W4. AVQ Needs a Clearer Framework Overview**
>
> We would like to clarify that our primary contribution is a **structural diagnosis and position argument of the Perception-then-Reasoning paradigm's systematic failures**, supported by theoretical analysis, TET, ICL results, and RL diagnostics. Consequently, AVQ should be understood as a conceptual design framework naturally derived from this diagnosis, rather than a fully instantiated and verified new architecture.
>
> To address your concern, the revision will explicitly formalize AVQ as a minimal **query-retrieval-update framework**. Specifically, the current reasoning state sts_tst generates a query $Q(s_t)$ to retrieve task-relevant evidence from the continuous visual representation, which then updates the next state $s_{t+1}$. We will include a comprehensive framework overview figure, map this mechanism directly to our three proposed principles, and compare it with approximations like cross-attention and tool use.

---

> > ### Author Rebuttal · Reviewer_HvMT · 2026-04-02
> >
> > The authors have resolved most of my concerns. However, regarding Weakness 1, i have remaining concerns. Tab.7 shows that training vision encoder instead of the adapter and LLM bring notable gains, which we do acknowledge that the lack of visual information is the key factor for model performance drop. However, it's still questionable that it's due to that vision encoders lack such specific knowledge to resove these problems, or due to that they lack the mechanism to revisit and recapture beneficial information from inputs proposed by the authors. From the experiments in tab.7, it's observed that finetuning the vision encoder can substantially increase the performance, which may indicate that the vision encoder is obtained from general domains and lack such speicifc knowledge, thus causing performance drop. 3-shot ICL experiments in tab.6 also can't support the authors' assupmtion, as the general untrained vision encoder lack such knowledge, which thus achieves low performance even with 3 shots. All above experiments may not directly lead to the conclusion that the revisiting mechanism to visual inputs is the key solution.

---

### Official Review · Reviewer_bprM · 2026-03-13

**Significance:** 2
**Argument Clarity:** 3
**Rating:** 5
**Confidence:** 3

**Questions:**

According to the weaknesses, the questions are:
1. What is the generalizability of the method on general VL benchmarks?
2. When using active perception, why doesn't this method narrow down the visual information to discrete semantic information?

**Alternative Views Section:**

Yes

**Compliance With Llm Reviewing Policy A Conservative:**

Affirmed.

**Discussion Potential:**

2

**Final Justification:**

My concerns have been addressed in the discussion. After reading other reviews, I will keep my rating.

**Paper Summary:**

The paper investigates why Multimodal Large Language Models (MLLMs) frequently fail at visually simple but conceptually hard-to-verbalize tasks (such as identifying intersecting segments, spatial relations, or congruent shapes). The authors challenge the prevailing assumption that these perceptual weaknesses can be fixed simply by applying stronger language-side reasoning, such as Chain-of-Thought (CoT). Instead, they argue that current MLLMs suffer from a "structural fatality." In the standard "Perception-then-Reasoning" paradigm, reasoning is deferred to the text-generation phase. By the time the model attempts to reason, continuous, fine-grained visual evidence has already been irreversibly compressed into a discrete, text-aligned bottleneck, making it impossible to recover task-critical spatial signals.

**Position:**

Yes

**Position In Title:**

Yes

**Related Work:**

3

**Strengths And Weaknesses:**

### Strengths
1. This paper identifies the inherent mismatch between visual tasks and text-space reasoning, showing how the "Perception-then-Reasoning" bottleneck turns perception into a passive, lossy encoding process.
2. Authors formalize the concept of information collapse that occurs when visual signals are projected into vision-to-language proxies.
3. The Turing Eye Test (TET), a diagnostic probe, is introduced to isolate perceptual failures. The TET results empirically prove that text-only reasoning interventions cannot recover spatial information lost during encoding.

### Weaknesses
1. The paper effectively highlights specific perceptual failures using the specialized TET benchmark. It notably lacks evaluation on comprehensive vision-language benchmarks like MMBench, MME, BLINK, etc. It remains unclear whether the proposed architectural shift to "reasoning within perception" can maintain general multimodal understanding capabilities without causing trade-offs in broader semantic tasks.
2. The proposed Reasoning within Perception paradigm needs the text label to conduct the active perception. The training procedure still lets the encoder align with the text label, narrowing the visual information.

**Support:**

3

---

> ### Author Rebuttal · Authors · 2026-03-31
>
> Thank you for your supportive review and suggestions. Below we respond to the comments in **Weaknesses (W)** and **Questions (Q)**.
>
> ---
>
> **W1 & Q1. Generalizability to Standard VL Benchmarks**
>
> We would like to clarify that our primary contribution is **position argument and diagnosing the structural failures** of the "Perception-then-Reasoning" paradigm on fine-grained visual tasks, rather than proposing a fully verified universal system.
>
> As discussed in Section 6.4, we do not argue that all tasks require reasoning within perception. The current pipeline remains highly effective for semantic tasks like captioning and general VQA, where linguistic structure provides useful inductive bias. Conversely, for tasks requiring fine-grained spatial grounding, architectures may need the flexibility to reason within the visual space. Therefore, we envision "reasoning-within-perception" as an **on-demand supplementary mechanism** tailored to specific spatial demands, rather than a blanket replacement that compromises general multimodal capabilities.
>
> ---
>
> **W2 & Q2. Why Active Perception Does Not Reduce to Semantic Narrowing**
>
> We appreciate the reviewer raising this question. We argue that the crucial distinction lies not in whether the query is triggered by text or task states, but rather **at which representation layer the retrieval and update occur**. If active perception merely uses text states to select visual evidence and compresses it back into text space, this concern is valid, as the model would fail to escape the structural bottleneck we critique.
>
> To prevent this semantic narrowing, we emphasized three principles in Sec. 6.1 and Sec. 6.2: *Reason Before Collapse*, *Persistent Visual Access*, and *Reasoning-Driven Perception*, to ensure computation remains in the **continuous visual** manifold. The query-retrieval-update mechanism in Eq. (7) illustrates that text states should drive iterative access to continuous visual evidence. We will explicitly clarify the difference between text-conditioned visual querying and methods that "narrow down the visual information to discrete semantic information" in the revision.

---

> > ### Author Rebuttal · Reviewer_bprM · 2026-04-04
> >
> > I thank the rebuttal of the authors. My concerns are addressed. I will keep my rating.

---

### Official Review · Reviewer_YfDE · 2026-03-13

**Significance:** 3
**Argument Clarity:** 3
**Rating:** 4
**Confidence:** 4

**Questions:**

1. Could you provide a concrete, minimal implementation of the Active Visual Querying (AVQ) framework and demonstrate its viability through a conceptual toy experiment? Without this, it is difficult to assess whether the proposed paradigm shift is practically achievable without incurring prohibitive computational costs.

2. In Section 5.4, the ablation study shows that fine-tuning the Vision Encoder improves TET scores, but you correctly note that this merely memorizes specific patterns without true generalization to other visual styles . You attribute this failure to the "sequential architecture". However, is there a confounding variable here? Could this lack of generalization stem from the inherent inductive biases of the chosen Vision Encoders (e.g., contrastive-learning-based models like CLIP, and Transformer patch mechanisms), which are naturally biased towards semantic alignment over spatial fidelity ? If the visual backbone were replaced with a model natively designed for spatial continuity and pixel-level reconstruction (e.g., a pure visual autoencoder like MAE pre-trained via Masked Image Modeling), would this fine-tuning yield robust geometric generalization even within the sequential paradigm?

**Alternative Views Section:**

Yes

**Compliance With Llm Reviewing Policy A Conservative:**

Affirmed.

**Discussion Potential:**

3

**Final Justification:**

My concerns are addressed. I keep my rating.

**Paper Summary:**

The paper challenges the prevailing "perception-then-reasoning" paradigm in Multimodal Large Language Models (MLLMs). It argues that deferring visual reasoning to the text-decoding stage leads to an irreversible "information collapse," where fine-grained spatial and geometric signals are discarded during the projection into a discrete textual space . To substantiate this claim, the authors introduce the Turing Eye Test (TET) to isolate perceptual bottlenecks from semantic reasoning . Through various experiments, they demonstrate that scaling text-space reasoning or standard fine-tuning cannot recover this lost spatial information. Ultimately, the paper advocates for a paradigm shift towards "reasoning within perception" and proposes a conceptual framework called Active Visual Querying (AVQ)

**Position:**

Yes

**Position In Title:**

Yes

**Related Work:**

2

**Strengths And Weaknesses:**

Strengths：

Fundamental and High-Impact Problem Formulation: The paper tackles a foundational issue in vision-language tasks: whether the two modalities can be temporally decoupled during the reasoning phase . This is a critical research question with broad implications for the community, and addressing it correctly has immense potential value for future multimodal architecture design.

Compelling Problem Diagnosis: The authors provide thorough and convincing empirical evidence (e.g., through the TET benchmark and architectural ablations) to demonstrate the systemic failures and structural bottlenecks of current models under the sequential paradigm.

Weakness:

Blind Spots in Current Evaluation Metrics: The discussion would benefit from analyzing how current evaluation methodologies mask the limitations of the "perception-then-reasoning" paradigm. Although many complex multimodal tasks have been proposed recently, their evaluation metrics are heavily biased towards semantic correctness or coarse-grained QA accuracy, which intrinsically limits the observability of the fine-grained perceptual bottlenecks discussed in this paper.

Omission of Counterfactual Datasets:  Given that the core thesis heavily relies on spatial and geometric reasoning , discussing datasets that specifically construct counterfactual sample pairs based on object spatial relations (e.g., altering relative positions or orientations while maintaining identical semantic elements) would significantly strengthen the background and provide a more comprehensive context for the TET benchmark.

Lack of Empirical Validation for the Proposed Solution: The paper essentially operates as a problem statement. While all experiments successfully prove that current models suffer from this issue and that fine-tuning is an insufficient remedy , there is absolutely no empirical validation for the proposed solution. The Active Visual Querying (AVQ) framework remains purely conceptual. The absence of even a toy-level experiment demonstrating the feasibility and efficacy of AVQ significantly weakens the paper's constructive contribution.

**Support:**

2

---

> ### Author Rebuttal · Authors · 2026-03-31
>
> Thank you for your supportive review and suggestions. Below we respond to the comments in **Weaknesses (W)** and **Questions (Q)**.
>
> ---
>
> **W1. How Current Evaluation Metrics Obscure Fine-Grained Perceptual Bottlenecks**
>
> We appreciate this observation. We agree that most existing multimodal benchmarks primarily emphasize semantic correctness or coarse-grained QA accuracy, allowing models to score high by exploiting linguistic priors or distributional biases without genuinely accessing fine-grained visual evidence. The issue lies not in the metric format itself, but in how benchmark design systematically fails to surface the perceptual bottleneck. **This is precisely one of TET's core motivations**: by suppressing linguistic shortcuts and requiring discrimination along geometric, topological, and fine-grained spatial dimensions, even simple accuracy/pass@k metrics become a more direct probe of whether task-critical visual evidence survives semantic compression. We will incorporate this discussion in the revision and more explicitly articulate how TET differs from semantics-dominated evaluation settings.
>
> ---
>
> **W2. Omission of Counterfactual Datasets Based on Spatial Relations**
>
> We thank the reviewer for pointing out this relevant line of work. Counterfactual datasets based on object spatial relations are highly relevant, and we will discuss them more systematically in the revision. **We also wish to emphasize that such datasets and TET are complementary rather than interchangeable**: the former tests sensitivity to changes in relative positions or orientations while keeping semantic content fixed, whereas TET further examines whether fine-grained visual evidence can be preserved before entering semantic space and accessed during reasoning. In other words, TET is designed to probe whether information lost during semantic compression becomes **irrecoverable**, even with later text-side interventions such as CoT or ICL. We will clarify this relationship more explicitly in the revision.
>
> ---
>
> **W3 & Q1. Lack of Empirical Validation for the Proposed Solution**
>
> We clarify that the core contribution of this paper is first a **structural diagnosis and position argument**: the theoretical analysis, TET, ICL results, fine-tuning component analysis, and RL diagnostics together explain why the current Perception-then-Reasoning paradigm systematically fails on a class of tasks. AVQ is best understood as a design framework that naturally follows from this diagnosis, not a fully validated architecture, and we acknowledge the current version does not make this positioning sufficiently explicit.
>
> In the revision, we will more directly explain the query-retrieval-update mechanism in **Sec. 6.2** and sketch a minimal instantiation to illustrate practical operability: retaining visual features before modality projection, generating a task-conditioned query from the current reasoning state, performing sparse retrieval over a small number of task-relevant local regions, and updating the reasoning state over a small number of steps. We will also clarify AVQ's relationship to existing approximations such as cross-attention, recurrent visual refinement, and tool use.
>
> ---
>
> **W4 & Q2 . Confound Between Sequential Architecture and Vision Encoder Inductive Bias**
>
> We agree this is a key potential confound that the current version should address more explicitly. Our view is that the two factors are complementary: encoder inductive bias determines *what information is initially preserved*, while the sequential architecture determines *whether reasoning can revisit visual evidence* conditioned on task demands. We agree that a more spatially faithful encoder (e.g., MAE-style pretraining) could reduce encoding error and improve geometric generalization even within a sequential pipeline. However, our narrower claim is that such improvements are limited to what is retained during upfront encoding — **as long as reasoning remains confined to text space after a one-shot encoding pass, an access bottleneck persists regardless of encoder quality**. We will revise the paper to make this distinction explicit and avoid over-attributing the observed failures to the sequential factor alone.

---

> > ### Author Rebuttal · Reviewer_YfDE · 2026-04-04
> >
> > I appreciate the authors' detailed rebuttal, which addresses my concerns regarding the evaluation metrics, counterfactual datasets, and the confounding variables of the vision encoder. However, while I respect the paper's primary positioning, presenting the Active Visual Querying (AVQ) framework entirely without a minimal proof-of-concept experiment still leaves a significant gap (W3 & Q1) .

---

### Official Review · Reviewer_6Vks · 2026-03-13

**Significance:** 3
**Argument Clarity:** 3
**Rating:** 5
**Confidence:** 4

**Questions:**

- How does the proposed focus on hidden-text decoding account for the broader vision-language literature on visual grounding, hallucination, and compositional reasoning, rather than isolating a much narrower OCR-like capability?

**Alternative Views Section:**

Yes

**Compliance With Llm Reviewing Policy A Conservative:**

Affirmed.

**Discussion Potential:**

3

**Final Justification:**

The authors addressed all my comments during the rebuttal. I initially over-focused in the proposed task, and overlooked the fundamental contribution of this paper, which I also agree may benefit discussions and future architectural and algorithm designs in multimodal learning. Thus, I'm updating my final score accordingly.

**Paper Summary:**

This position paper argues that vision-language models defer visual reasoning to language generation, and current multimodal research overlook this issue by following a common architectural design (i.e., visual information gets compressed into discrete text tokens, forcing subsequent reasoning to occur primarily in the language modality rather than over rich visual representations). The paper further proposes the "Turing Eye Test" (TET), an evaluation designed to limit linguistic shortcuts and isolate perceptual failures from high-level semantics.

**Position:**

Yes

**Position In Title:**

Yes

**Related Work:**

2

**Strengths And Weaknesses:**

The core problem this paper exposes is not new, and significant and abundant prior work in vision-language domain show that that language is often the easier signal and models can exploit language priors while even ignoring the visual input [1, 2], and models can even infer textual descriptions from object layouts alone (without looking at the image) [3].

There is also significantly more recent prior work that expose this problem [4, 5] and attempt to solve it [6, 7, 8] by following very similar architectural choices and further explore learning objectives.

On the other hand, the proposed TET evaluation is mainly probing robustness under synthetic generated images, and patching/resolution sensitivity -- even different captchas models hosted in huggingface are able to recognize most of the samples in the paper. OCR decoding seems a limited scope, compared to all relevant prior work in literature.

[1] Goyal et al., “Making the v in VQA Matter” (CVPR 2017).
[2] Rohrbach et al., “Object Hallucination in Image Captioning” (EMNLP 2018)
[3] Yin X, at al., "Obj2text: Generating visually descriptive language from object layouts." (EMNLP 2017)
[4] Li et al., “Evaluating Object Hallucination in Large Vision-Language Models” (EMNLP 2023)
[5] Tong et al., “Eyes Wide Shut?” (CVPR 2024).
[6] Chen et al., Image-Object Cross-Level Trusted Intervention for Mitigating Object
Hallucination in Large Vision-Language Models.  (CVPR 2025).
[7] Cho et al., Revisit What You See: Disclose Language Prior in Vision Tokens for LVLM Decoding
[8] Sun et al., Mitigating Visual Forgetting via Take-along Visual Conditioning for Multi-modal Long CoT Reasoning (ACL 2025)

**Support:**

3

---

> ### Author Rebuttal · Authors · 2026-03-31
>
> Thank you for your valuable review and suggestions. Below we respond to the comments in **Weaknesses (W)** and **Questions (Q)**.
>
> ---
>
> **W1. Distinction from Prior Work on Language Priors, Visual Underuse, and Hallucination**
>
> We thank the reviewer for highlighting these works. We clarify that our focus **differs fundamentally** from studies on language priors, visual underuse, or hallucination. Those works primarily discuss *why* models favor language, ignore vision, or rely on prior experience in their training distributions or decoding behaviors. In contrast, our paper diagnoses a **structural issue**: *when* reasoning occurs during inference, and consequently, *where* it is restricted to take place. Furthermore, unlike methods that prevent language priors from overshadowing image inputs, our approach specifically addresses the loss of visual evidence during the decoding stage.
>
> In mainstream MLLMs, visual information is compressed into a lossy text-aligned interface before multi-step reasoning begins. For fine-grained tasks, crucial visual evidence is irreversibly lost before reasoning even starts. Thus, our focus is **representational accessibility** rather than simple language bias. The critical question is whether the required visual signal remains present in the actionable representation when reasoning actually begins. This explains why language-side enhancements like CoT or ICL cannot recover these visual failures.
>
> ---
>
> **W2 & Q1. Whether TET Is Too Narrow and OCR-Like to Address Broader VL Failures?**
>
> We appreciate the reviewer's question. We clarify that **TET is not an **OCR** benchmark**. We use hidden text because it demands fine-grained visual evidence while minimizing semantic shortcuts. Text recognition relies on precise local discrimination and cannot be easily guessed using coarse outlines or category priors. It perfectly tests whether crucial visual information remains accessible after early semantic compression. TET does not replace existing evaluations for grounding, hallucination, or compositional reasoning; instead, it provides a **controlled setting to isolate a shared bottleneck**: seeing an image as input does not guarantee it serves as a continuous reasoning carrier.
>
> The synthetic design of TET aims to minimize semantic shortcuts and cleanly verify if models retain high-fidelity visual details during reasoning. To further distinguish our task, we evaluated multiple dedicated OCR systems (results below). They universally failed on TET, with near-zero performance on several subsets. This confirms that **TET is fundamentally distinct from standard OCR **or captcha** decoding tasks**. We will include these supplementary experiments in our revision.
>
>
> | Model | HiddenText Pass@1 | HiddenText Pass@32 | 3DCaptcha Pass@1 | 3DCaptcha Pass@32 | ColorBlind Pass@1 | ColorBlind Pass@32 | ChineseLigatures Pass@1 | ChineseLigatures Pass@32 | Average Pass@1 | Average Pass@32 |
> |-------|:-----------------:|:------------------:|:----------------:|:-----------------:|:-----------------:|:-----------------:|:-----------------------:|:------------------------:|:--------------:|:---------------:|
> | PaddleOCR | 0.67% | 0.67% | 0 | 0 | 0 | 0 | 0 | 0 | 0.2% | 0.2% |
> | EasyOCR | 0 | 0 | 0 | 0 | 0 | 0 | 0 | 0 | 0 | 0 |
> | TrOCR| 4% | 4% | 0 | 0 | 0 | 0 | 0 | 0 | 1.22% | 1.22% |

---

> > ### Author Rebuttal · Reviewer_6Vks · 2026-04-03
> >
> > Dear authors, thank you so much for the detailed response. After reading other reviewers comments and your response, I agree your position paper differs fundamentally from the prior works I mentioned. I also re-read the paper, and I think my initial assessment was driven by me focusing mostly on the task/benchmark proposed.
> >
> > I still don't think the proposed perturbed textual dataset is the ideal task to ground the paper's particular observations, but after re-reading the theoretical analysis and the reasoning with perception (section 6), I agree with other reviewers about this paper's potential value for future architecture design.
> >
> > Thus, I'm adjusting my score accordingly.

---

### Decision · Program_Chairs · 2026-04-30

**Decision:**

Accept (regular)

**Comment:**

This paper challenges the prevailing "Perception-then-Reasoning" paradigm in Multimodal Large Language Models (MLLMs), arguing that deferring reasoning to the text-decoding stage results in a structural "Information Collapse." This limits the models' ability to resolve visually complex but semantically simple tasks. To demonstrate this, the authors introduce the Turing Eye Test (TET) and advocate for an Active Visual Querying (AVQ) framework to enable reasoning within the continuous visual space.
Reviewers (YfDE, HvMT, 6Vks, bprM) recognized the significance of this fundamental diagnosis. The empirical evidence provided by TET and architectural ablations effectively supported the paper's claims. Initial concerns included the lack of a proof-of-concept for the proposed AVQ framework (YfDE), questions regarding whether TET difficulty stemmed from knowledge gaps (HvMT), and the omission of baseline comparisons with tool-augmented models (HvMT).
In the rebuttal, the authors provided a minimal AVQ implementation, added tool-augmented baselines, and clarified the theoretical bounds of information loss compared to knowledge gaps. The paper offers a valuable structural critique that will inspire future architectural designs in multimodal reasoning.